# Characterisation and induction of tissue-resident gamma delta T-cells to target hepatocellular carcinoma

Nekisa Zakeri [1], Andrew Hall[2], Leo Swadling [1], Laura J. Pallett [1], Nathalie M. Schmidt [1], Mariana O. Diniz[1], Stephanie Kucykowicz[1], Oliver E. Amin [1], Amir Gander[3], Massimo Pinzani[2], Brian R. Davidson[3], Alberto Quaglia[4] & Mala K. Maini [1]✉

Immunotherapy is now the standard of care for advanced hepatocellular carcinoma (HCC), yet many patients fail to respond. A major unmet goal is the boosting of T-cells with both strong HCC reactivity and the protective advantages of tissue-resident memory T-cells ($T_{RM}$). Here, we show that higher intratumoural frequencies of γδ T-cells, which have potential for HLA-unrestricted tumour reactivity, associate with enhanced HCC patient survival. We demonstrate that γδ T-cells exhibit bona fide tissue-residency in human liver and HCC, with γδ$T_{RM}$ showing no egress from hepatic vasculature, persistence for >10 years and superior anti-tumour cytokine production. The Vγ9Vδ2 T-cell subset is selectively depleted in HCC but can efficiently target HCC cell lines sensitised to accumulate isopentenyl-pyrophosphate by the aminobisphosphonate Zoledronic acid. Aminobisphosphonate-based expansion of peripheral Vγ9Vδ2 T-cells recapitulates a $T_{RM}$ phenotype and boosts cytotoxic potential. Thus, our data suggest more universally effective HCC immunotherapy may be achieved by combining aminobisphosphonates to induce Vγ9Vδ2$T_{RM}$ capable of replenishing the depleted pool, with additional intratumoural delivery to sensitise HCC to Vγ9Vδ2$T_{RM}$-based targeting.

---

[1] Division of Infection & Immunity, Institute of Immunity & Transplantation, University College London, London, UK. [2] Institute for Liver & Digestive Health, Royal Free London NHS Foundation Trust, London, UK. [3] Division of Surgery, University College London, London, UK. [4] Department of Cellular Pathology, Royal Free London NHS Foundation Trust and UCL Cancer Institute, London, UK. ✉email: m.maini@ucl.ac.uk

  1

Hepatocellular carcinoma (HCC) is the third most common cause of cancer-related death worldwide[1,2], often presenting at an advanced stage when treatment options are limited. Immunotherapy aims to induce T-cells with efficient tumour targeting and durable immune surveillance capacity. The potential for immunotherapy to reduce the high mortality of HCC has been exemplified by the response to the combination of anti-programmed death ligand-1 (PD-L1) and anti-vascular endothelial growth factor (VEGF), which has now become standard-of-care in advanced disease; however, this combination still only achieves responses in around one third of patients[3,4], underscoring the urgent need for additional immunotherapeutic approaches that can deliver more consistent and sustained responses. A major reason for the failure of existing checkpoint inhibitors is the scarcity of T-cells capable of recognising expressed tumour antigens in many individuals, such that insufficient tumour-specific responses are available for boosting[5–7]. A further limitation of existing immunotherapies is that endogenously boosted or adoptively transferred T-cells may fail to infiltrate and survive within the tumour and/or maintain functionality in this immunosuppressive environment[2,4,8]. Here, we present data supporting a novel strategy for expanding γδ T-cells with the potential to lyse HCC and to persist long-term, retaining functionality within the tumour niche.

γδ T-cells can play a critical role in tumour immunosurveillance, exemplified by the favourable prognostic signature of tumour-infiltrating γδ T-cells across many human cancer types[9–13], and an increased susceptibility to cancer in γδ T-cell deficient mice[14–16]. γδ T-cells are attractive effector cells for cancer immunotherapy, due to their MHC-unrestricted antigen recognition and lack of dependence on cancer neo-antigens[17,18]. The main subtypes in humans, Vδ1 and Vδ2 T-cells, both recognise stress-induced molecules including MHC-Class 1 chain-related protein A/B and UL16-binding proteins, expressed at variable levels on tumour cells[11,17–20]. Additionally, the predominant γδ T-cell subset in blood, Vγ9Vδ2 T-cells, can easily be expanded on a clinical scale for adoptive cell transfer to target phosphorylated intermediates of the isoprenoid biosynthesis pathway (isopentenyl pyrophosphate, IPP), upregulated in transformed cells and presented via butyrophilin 2A1 and 3A1 transmembrane proteins[20,21]. Early phase clinical trials have demonstrated adoptive cell transfer of in vitro expanded blood Vγ9Vδ2 T-cells to be safe and well-tolerated in several cancer types[18,22–24], but their overall efficacy remains low.

Accumulating evidence suggests that exploiting features of tissue-resident memory T-cells ($T_{RM}$) may help to uncover novel strategies for more effective cancer immunotherapies[25–27]. $T_{RM}$ are phenotypically and functionally distinct sentinels in the liver, capable of long-lived retention, and well positioned for rapid and potent front-line immunosurveillance[28–30]. αβ $T_{RM}$ are marked by the expression of the tissue retention molecules CD69 (a negative regulator of sphingosine-1-phosphate-mediated T cell egress), and the integrins CD103 (binding E-cadherin on epithelial cells) and/or CD49a (binding collagen IV)[25–27]. The control of tumours has been shown to be highly dependent on αβ $T_{RM}$ in murine models and human studies of various cancers including HCC[27,31–40], but the contribution of γδ $T_{RM}$ has not yet been investigated in HCC. Although most studies of T-cell tissue-residence have focused on αβCD8+, the Vδ1 subset of γδ T-cells are generally considered to be tissue-resident, supported by recent data confirming expression of tissue retention/homing markers and distinct T cell receptor (TCR) clones by Vδ1 T-cells in human liver[41]. By contrast, Vγ9Vδ2 T-cells are classically regarded as the circulating γδ T-cell subtype, and have yet to be shown to be capable of acquiring a functionally intact $T_{RM}$ phenotype in the liver or HCC.

In this study, we find that γδ T-cell infiltration of HCC associates with a favourable prognosis, assessed both by tumour size and patient survival. We demonstrate that Vγ9Vδ2 T-cells are selectively reduced in frequency in the blood, livers, and tumours of patients with HCC, but display the capacity to acquire an intratumoural γδ $T_{RM}$ (CD69+CD49a+ or CD69+CD103+) phenotype. Intratumoural γδ T-cells with a $T_{RM}$ phenotype show enhanced capacity to maintain the production of cytokines favouring their survival and their anti-tumour potential, in the liver and within HCC tumour-infiltrating lymphocytes (TIL). The bona fide nature of the γδ $T_{RM}$ phenotype is confirmed by a lack of hepatic vasculature egress and by persistence of CD69+CD49a+ γδ $T_{RM}$ progeny for more than a decade in human leukocyte antigen (HLA)-mismatched liver allografts. Therefore, we develop a rationale for use of the aminobisphosphonate Zoledronic acid (ZOL) to optimise the anti-tumour efficacy of Vγ9Vδ2 T-cells that can become locally resident; we show that its local application to HCC cell lines allows IPP accumulation for specific lysis by liver/HCC-infiltrating Vγ9Vδ2 T-cells, whilst its use for in vitro clinical-scale expansion of peripheral Vγ9Vδ2 T-cells induces a liver-homing $T_{RM}$ profile.

## Results

**Compartmentalisation of Vδ1 and Vδ2 T-cells with a tissue-resident phenotype in human liver.** To examine the potential for the two major subsets of human γδ T-cells, Vδ1 and Vδ2, to acquire tissue-residence, we first compared their relative abundance in human liver (tumour-free liver tissue from surgical resections or explants) to paired peripheral blood samples (gating strategy Supplementary Fig. 1a). Vδ1 T-cells are regarded as prototypic tissue γδ T-cells and were accordingly significantly enriched in the liver compared to their very low frequencies in matched blood samples (Supplementary Fig. 1b). Vδ2 T-cells, whilst more frequent in blood than Vδ1, were also detectable in the liver, comprising up to 5% of intrahepatic CD45+ leukocytes (Supplementary Fig 1b). Intrahepatic Vδ2 T-cells consisted primarily of Vγ9+Vδ2 T-cells (henceforth referred to as Vγ9Vδ2), with a very low frequency of the recently described Vγ9−Vδ2 sub-population[42] (Supplementary Fig. 1c).

We next investigated if Vδ1 and Vγ9Vδ2 T-cells could express a tissue-resident phenotype in human liver, assessing the prototypic markers CD69, CD103, and CD49a. We found that subpopulations of both Vδ1 and Vγ9Vδ2 T-cells isolated from human liver expressed combinations of tissue-retention markers characteristic of $T_{RM}$, that could not be detected in paired peripheral blood (Fig. 1a, b). Vδ1 and Vγ9Vδ2 T-cells co-expressing the tissue retention molecules CD69 with CD103 or CD49a were compartmentalised in the liver and excluded from blood (Fig. 1a, b). These tissue-resident γδ T-cell subsets were primarily composed of CD27−CD45RA− effector memory cells (Supplementary Fig. 1d), and consisted of distinct CD69+CD103+ or CD69+CD49a+ subsets, with a smaller sub-population expressing all three tissue-retention markers (Fig. 1c). CD69+CD49a+ co-expression was more prevalent than CD69+CD103+ on intrahepatic Vδ1 and Vγ9Vδ2 T-cells (Fig. 1c). The proportion of intrahepatic T-cells expressing CD69+CD49a+ showed a strikingly similar range within the cohort for Vδ1, Vγ9Vδ2, and αβCD8+ T-cells (mean 22.2 ± 18.6%, 21.1 ± 15.2%, 20.1 ± 12.3%, respectively, Supplementary Fig. 1e), with a significant correlation between expression levels on Vγ9Vδ2 and αβCD8+ T-cells within individuals (Supplementary Fig. 1e), suggesting shared determinants of these residency markers. The percentage of CD69+CD103+ expression on intrahepatic Vδ1 and Vγ9Vδ2 T-cells demonstrated a weak inverse correlation with patient age (Supplementary Fig. 1f). No

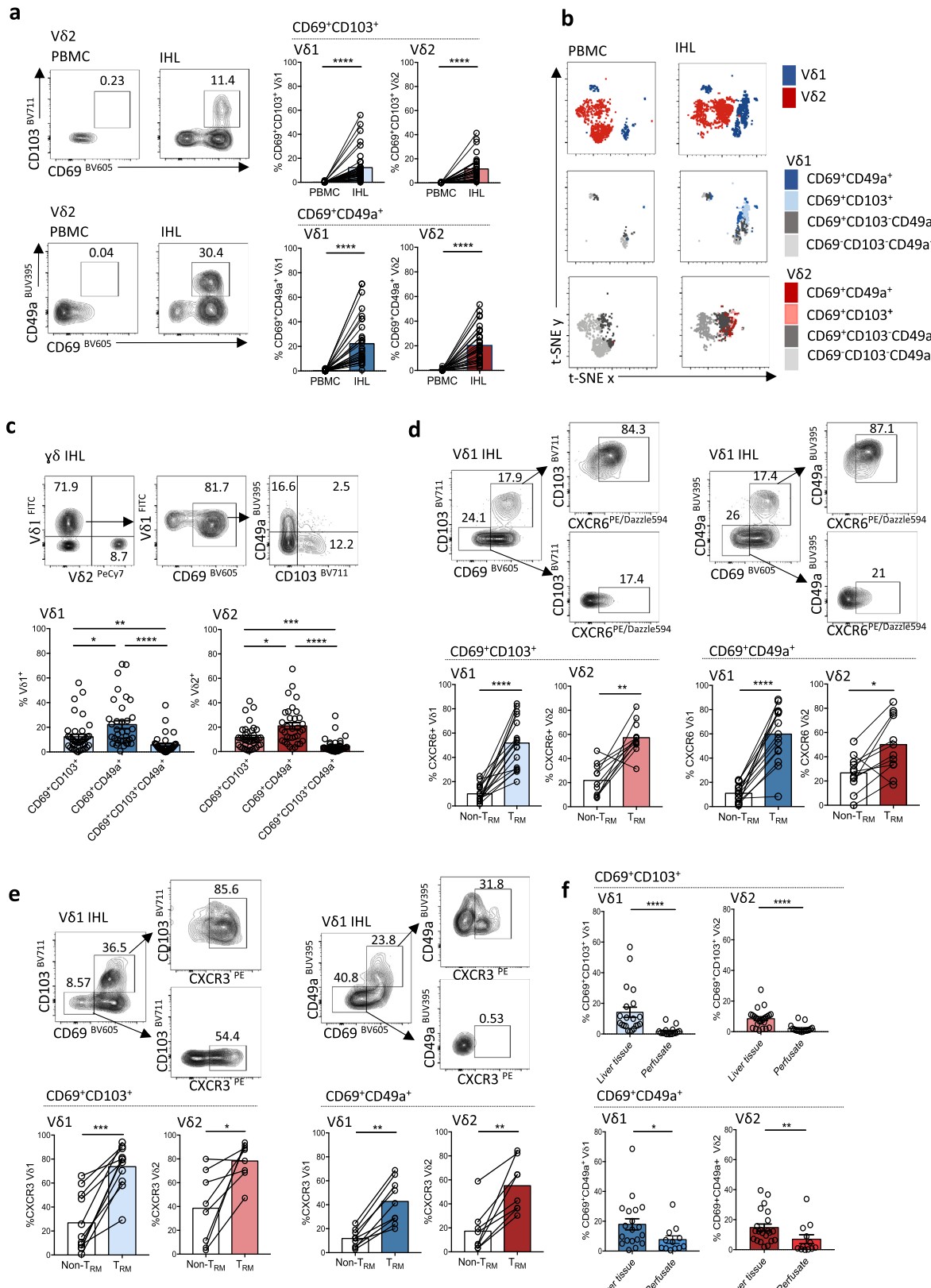

sex related differences in Vδ1 and Vγ9Vδ2 $T_{RM}$ expression were observed (Supplementary Fig. 1f).

To further assess Vδ1 and Vγ9Vδ2 T-cells tissue-homing capability, we measured expression of the prototypic liver chemokine receptors CXCR6 and CXCR3. On ex vivo staining of bulk intrahepatic lymphocytes (IHL), we found that both Vδ1

and Vγ9Vδ2 $T_{RM}$ (CD69+CD49a+ or CD69+CD103+) showed increased expression of CXCR6 and CXCR3 in comparison to their non-$T_{RM}$ counterparts (CD69−CD103− or CD69−CD49a−; Fig. 1d, e). Ligands for these chemokine receptors are expressed by hepatic sinusoidal endothelium, promoting homing, tethering and/or transmigration into the parenchyma[43,44]. Consistent with

**Fig. 1 Compartmentalisation of Vδ1 and Vγ9Vδ2 T-cells with a tissue-resident phenotype in human liver. a** Representative flow cytometry plots and summary data of CD69+CD103+ or CD69+CD49a+ co-expression on Vδ1 and Vγ9Vδ2 T-cells from intrahepatic lymphocytes (IHL) of tumour-free liver tissue compared to paired peripheral blood mononuclear cells (PBMC) ($n = 35$–$44$; $p < 0.0001$). **b** t-distributed stochastic neighbour embedding (t-SNE) was applied to flow cytometry expression data (concatenated PBMC $n = 3$ and IHL $n = 5$) of Vδ1 (blue) and Vγ9Vδ2 (red) T cells (top row); plots coloured by CD69, CD49a and CD103 expression on Vδ1 T-cells (middle row) and Vγ9Vδ2 T-cells (bottom row). **c** Frequencies of CD69+CD103+, CD69+CD49a+ and CD69+CD49a+CD103+ intrahepatic Vδ1 and Vγ9Vδ2 T-cells ($n = 40$; Vδ1 $p = 0.02$, $p = 0.004$, $p < 0.0001$; Vγ9Vδ2 $p = 0.02$, $p = 0.0008$; $p < 0.0001$). **d** Frequencies of CXCR6-expressing CD69+CD103+ or CD69+CD49a+ Vδ1 and Vγ9Vδ2 $T_{RM}$ ($n = 17$; Vδ1 $p < 0.0001$; Vγ9Vδ2 $p = 0.002$, $p = 0.02$). **e** Frequencies of CXCR3-expressing CD69+CD103+ ($n = 14$) or CD69+CD49a+ ($n = 9$–$11$) Vδ1 and Vγ9 Vδ2 $T_{RM}$ (Vδ1 $p = 0.001$, $p = 0.004$; Vγ9Vδ2 $p = 0.008$, $p = 0.02$). **f** Frequencies of Vδ1 and Vγ9Vδ2 $T_{RM}$ in IHL from healthy liver tissue (disease-free margins of CRCLM) ($n = 24$) compared to healthy donor liver transplant perfusates ($n = 17$, Vδ1 $p < 0.0001$, $p = 0.03$; Vγ9Vδ2 $p < 0.001$, $p = 0.0096$). Each symbol represents a study participant, with error bars showing the mean ± SEM (**a**, **c**–**f**); two-tailed $p$-values were determined using Wilcoxon matched-pairs signed rank test (**a**, **d**, **e**), Kruskal–Wallis test (ANOVA) followed by Dunn's post-hoc multiple comparisons test (**c**), Mann–Whitney test (**f**). *$p < 0.05$; **$p < 0.01$; ***$p < 0.001$; ****$p < 0.0001$.

firm tethering or transmigration beyond the vasculature, we observed that Vδ1 and Vγ9Vδ2 $T_{RM}$ were less likely to be flushed out of the liver in perfusates from healthy donor liver allografts than to be isolated from liver tissue digests (healthy background liver from resected colorectal cancer liver metastases, CRCLM) (Fig. 1f). However, the isolation of a clear population of CD49a expressing Vδ1 and Vγ9Vδ2 T-cells from perfusates (Fig. 1f), albeit at a lower frequency than in liver tissue, is in line with the demonstration that αβCD8+ $T_{RM}$ can patrol the liver sinusoidal vasculature[28,45].

In keeping with classical features of αβ T-cell tissue-residency, intrahepatic Vδ1 and Vγ9Vδ2 $T_{RM}$ showed reduced expression of the endothelial homing receptor CX3CR1, and lacked expression of the lymph node homing receptor CD62L (L-selectin) (Supplementary Fig. 1g, h). In addition, γδ $T_{RM}$ demonstrated a transcription factor profile characteristic of αβ $T_{RM}$ with higher Blimp-1 and lower Eomes expression than their non-$T_{RM}$ counterparts[28,46] (Supplementary Fig. 1i). Vδ1 and Vγ9Vδ2 $T_{RM}$ also had a trend towards higher expression of Tcf1 (Supplementary Fig. 1i), a transcription factor conferring stem cell-like longevity on T-cells[47].

**Long-lived hepatic retention and replenishment of Vδ1 and Vγ9Vδ2 $T_{RM}$.** The absence of Vδ1 and Vγ9Vδ2 T-cells with a tissue-resident phenotype (CD69+CD103+ or CD69+CD49a+) in peripheral venous blood points to an inability of these cells to egress from the liver. However, it is difficult to definitively conclude from this that they are completely compartmentalised in the liver, since γδ $T_{RM}$ leaving the liver vasculature at low frequencies would be difficult to detect following dilution in the peripheral circulation. We, therefore, sampled hepatic venous blood (draining directly from the liver) and portal venous blood (entering the liver), obtained from patients with cirrhosis undergoing transjugular intrahepatic portosystemic shunt procedures, in order to examine for potential low-level egress of γδ $T_{RM}$ from the liver or gut/spleen, respectively. γδ T-cell frequency appeared higher in hepatic venous blood compared to portal venous blood (Fig. 2a). γδ T-cells expressing CD69+CD103+ or CD69+CD49a+ could not be detected in hepatic venous blood or portal venous blood (Fig. 2b), providing further evidence to support the tissue-compartmentalisation of intrahepatic Vδ1 and Vγ9Vδ2 $T_{RM}$ subsets, with no low-level egress from the liver, gut or spleen.

In addition to tissue compartmentalisation, another feature of bona fide $T_{RM}$ is their long-lived retention within organs, critical for sustained immune surveillance of cancer. To investigate whether Vδ1 and Vγ9Vδ2 T-cells with a tissue-resident phenotype had this capacity, we tracked their longevity using donor and recipient HLA-mismatched liver allografts. Utilising HLA-specific monoclonal antibody staining, we were able to differentiate between the original donor-derived (liver-resident) γδ T-cells and the recipient-derived (blood-derived) γδ T-cells in two liver allografts, explanted 7 and 11 years following their initial transplantation (Fig. 2c, Supplementary Fig. 2a, patient characteristics in Supplementary Table 1). By both of the time points examined, the majority of the intrahepatic γδ T-cell population had been replaced with recipient-derived γδ T-cells (Fig. 2c). However, a small population of donor-derived γδ T-cells, constituting around 3% of the total intrahepatic γδ T-cell pool had persisted within the liver allografts, including in the explant obtained 11 years following transplantation (Fig. 2c). No donor-derived γδ T-cells were detected in the peripheral blood of the recipients, arguing against systemic chimerism accounting for intrahepatic persistence (Supplementary Fig. 2b). Of note, the proportion of Vδ1 and Vγ9Vδ2 T-cells was similar between the recipient-derived and donor-derived γδ T-cell populations, suggesting these equilibrate according to liver-specific factors (Fig. 2c).

The donor-derived intrahepatic γδ T-cell population persisting in liver allografts displayed a predominant CD69+CD49a+ phenotype, with little CD103 co-expression, indicating that CD49a expression, in particular, marked the subset of γδ T-cells capable of long-term retention and survival or renewal in the liver (pan-γδ in Fig. 2d, Vδ1 and Vγ9Vδ2 T-cell subsets in Supplementary Fig. 2c, d). Furthermore, the larger recipient-derived γδ T-cell fraction was capable of acquiring high CD69 expression, with a small proportion co-expressing CD49a or CD103 (Fig. 2d, Supplementary Fig. 2c, d). This demonstrates that recipient γδ T-cells infiltrating into the liver from the peripheral circulation were able to partially acquire a de novo $T_{RM}$ phenotype to replenish the intrahepatic γδ $T_{RM}$ cell pool.

Although limited by the scarce availability of these valuable samples, our data revealed the potential for a small population of long-lived or self-renewing CD69+CD49a+ γδ $T_{RM}$ to persist in the human liver for more than a decade, whilst being partially replenished from the peripheral circulation.

**Distinct functional profile of hepatic γδ $T_{RM}$.** To further assess the suitability of liver γδ $T_{RM}$ for immunotherapeutic targeting, we assessed their functional potential by ex vivo analysis of freshly isolated IHL. Both Vδ1 and Vγ9Vδ2 $T_{RM}$ (CD69+CD49a+ or CD69+CD103+) tended to express higher levels of the T-cell activation marker HLA-DR than their non-resident counterparts (Fig. 3a, Supplementary Fig. 3a). However, Vδ1 and Vγ9Vδ2 $T_{RM}$ displayed markedly reduced cytotoxic potential, demonstrated by lower expression of the serine protease granzyme B on direct ex vivo staining (Fig. 3b, Supplementary Fig. 3b). By contrast, both Vδ1 and Vγ9Vδ2 $T_{RM}$ were significantly more capable of rapid production of the pro-survival cytokine IL-2 than their non-$T_{RM}$ counterparts, with a striking mean of 65% and 74% of Vδ1 and

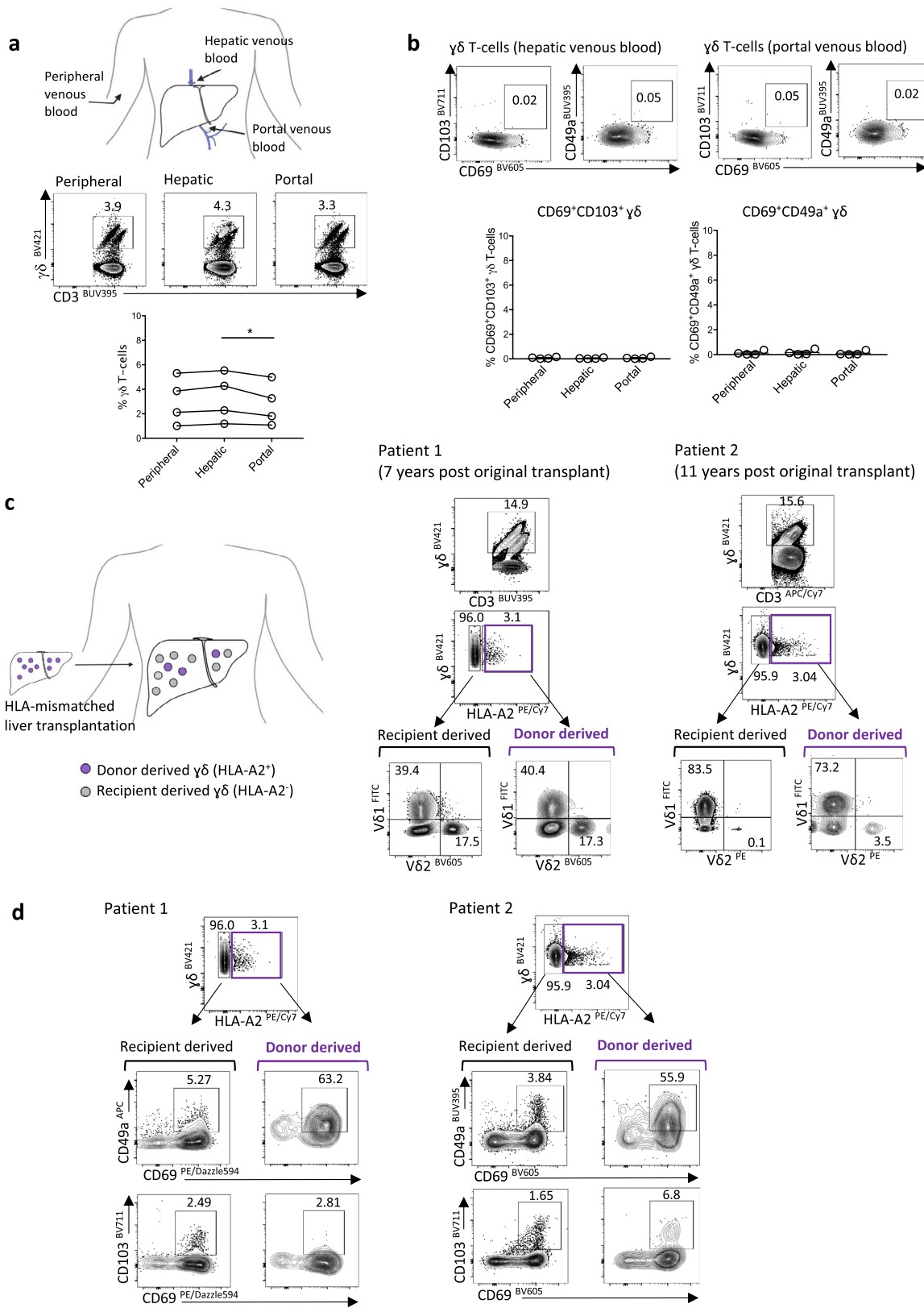

Vγ9Vδ2 T_RM, respectively, producing IL-2 after just 4 h of stimulation (Fig. 3c, Supplementary Fig. 3c). The majority of Vδ1 and Vγ9Vδ2 T_RM also produced the anti-tumour cytokine IFN-γ rapidly upon stimulation (Fig. 3d, Supplementary Fig. 3d). PD-1 expression was higher on Vδ1 and Vγ9Vδ2 T_RM compared to their non-T_RM counterparts ex vivo (Fig. 3e, Supplementary Fig.

3e); however, PD-1^high Vδ1 and Vγ9Vδ2 T-cells maintained their capacity to produce IFN-γ upon stimulation (Fig. 3f), congruent with the finding that PD-1-expressing αβCD8+ T_RM do not represent a functionally exhausted population[28,35]. Furthermore, no significant increase in intrahepatic Vγ9Vδ2 T-cell IFN-γ or Granzyme B expression was observed using anti-PD-L1 blockade

**Fig. 2 Long-lived hepatic retention and replenishment of Vδ1 and Vγ9Vδ2-T$_{RM}$. a** Representative flow cytometry plot and summary data of total γδ T-cells (as percentage of total CD3$^+$ T-cells) in peripheral, hepatic and portal venous blood obtained from cirrhotic patients undergoing transjugular intrahepatic portosystemic shunt procedures ($n = 4$, $p = 0.04$). **b** Representative flow cytometry plot and summary data demonstrating absence of CD69$^+$CD49a$^+$ or CD69$^+$CD103$^+$ γδT$_{RM}$ in peripheral, hepatic, portal venous blood ($n = 4$ matched samples). **c** Flow cytometry plot for the identification of donor human leukocyte antigen (HLA) A2$^+$ and recipient HLA A2$^-$ derived γδ T-cells from two explants obtained 7 years or 11 years following HLA-mismatched liver transplantation. Vδ1 and Vγ9Vδ2 T-cell frequencies within recipient-derived and donor-derived γδ T-cell subsets. **d** CD69$^+$CD49a$^+$ and CD69$^+$CD103$^+$ expression on donor-derived (HLA A2$^+$) and recipient-derived (HLA A2$^-$) γδ T-cell subsets ($n = 2$). Error bars, mean ± SEM (**b**). Two-tailed $p$-values were determined by Friedman test with Dunn's post-hoc test for multiple comparisons (**a**, **b**).

in combination with γδ-TCR stimulation in vitro (Supplementary Fig. 3f).

Taken together, assessing the functional potential of Vδ1 and Vγ9Vδ2 T$_{RM}$ isolated from the human liver directly ex vivo or after 4 h stimulation showed they are activated and skewed towards IL-2 and IFN-γ rather than cytotoxic effector function.

**γδ T-cell counts in HCC associate with tumour size and patient survival.** Having established a signature of γδ T-cells with classical tissue-residency features in human liver, we next investigated whether γδ T-cells could exert a role in HCC. First, we used immunostaining to compare the absolute numbers of γδ T-cells within HCC and paired background liver tissue from patients undergoing curative surgical resection ($n = 28$, patient characteristics, Supplementary Table 2). γδ T-cells could be visualised infiltrating HCC (Fig. 4a, Supplementary Fig. 4a) but their numbers were significantly reduced compared to paired tumour-free liver tissue (Fig. 4b). This trend was unchanged by the presence or absence of underlying cirrhosis (Supplementary Fig. 4b). The γδ/CD3$^+$ T-cell ratio was similarly reduced within HCC, due to preservation of total CD3$^+$ T-cell numbers between HCC and background liver tissue (Fig. 4b).

Importantly, higher intratumoural γδ T-cell numbers were associated with both smaller HCC tumour size at resection (maximum diameter ≤ 3 cm) (Fig. 4c), and enhanced 3-year overall patient survival following surgical resection (Fig. 4d). This finding was specific to intratumoural γδ T-cells, with no associations observed between total CD3$^+$ T-cell or γδ T-cell numbers within background liver and HCC tumour size or 3-year overall patient survival (Fig. 4c, d, Supplementary Fig. 4c–e). Consistent with this, analysis of the Cancer Genome Atlas database using the Gene Expression Profiling Interactive Analysis 2 (GEPIA2) web server[48], demonstrated a γδ-TCR gene signature (TRDC, TRGC1, TRGC2) to be significantly associated with overall survival in a larger cohort of HCC patients ($n = 364$), with higher Vγ9Vδ2 and non-Vγ9Vδ2 γδ-TCR gene expression signatures both associated with increased patient survival (Supplementary Fig. 4f, g)[48,49].

**Vγ9Vδ2 T-cells are selectively depleted, but can acquire tissue-residence, within HCC TILs.** To further define the depletion of γδ T-cells within HCC, we quantified Vδ1 and Vγ9Vδ2 T-cell subsets in blood, tumour-free liver and tumour tissue from patients with HCC compared to colorectal cancer liver metastases (CRCLM), by flow cytometry (patient characteristics, Supplementary Tables 3 and 4). No significant differences in the frequencies of Vδ1 T-cells were seen, but Vγ9Vδ2 T-cells were significantly reduced in the blood, liver and tumour of HCC compared to CRCLM patients (Fig. 5a). The reduction of Vγ9Vδ2 T-cells in HCC was likely attributable to the background liver cirrhosis in these donors, since we found a similar depletion in cirrhotic livers without HCC (Supplementary Fig. 5a), supported by recent literature describing increased activation and

subsequent apoptosis of liver sinusoidal Vγ9Vδ2 T-cells in cirrhosis[50].

Vδ1 T-cells had an increased proportion of terminally differentiated effector memory (TEMRA, CD27$^-$CD45RA$^+$) cells (Fig. 5b, Supplementary Fig. 5b), and demonstrated higher expression of the T-cell activation markers HLA-DR, CD38 and CD25 on direct ex vivo staining compared to Vγ9Vδ2 T-cells within HCC and paired tumour-free liver (Fig. 5c, Supplementary Fig. 5c). Despite their reduced frequency, Vγ9Vδ2 T-cells infiltrating background liver in HCC patients were better able to produce IFN-γ following 4 h stimulation than Vδ1 T-cells (Fig. 5d). Vγ9Vδ2 T-cells within HCC TILs also maintained high IFN-γ expression upon stimulation in the majority of HCC samples (Fig. 5d), although they had reduced granzyme B expression directly ex vivo (Fig. 5e).

In view of the long-lived retention of CD49a-expressing γδT$_{RM}$ we had demonstrated in the liver, we examined whether γδ T-cells infiltrating into HCC could acquire this residency profile. Both Vδ1 and Vγ9Vδ2 T-cells were able to acquire a T$_{RM}$ phenotype in HCC and CRCLM Fig. 5f, g). Co-expression of CD49a was higher than CD103 on intrahepatic and intratumoural Vδ1 and Vγ9Vδ2 T$_{RM}$ subsets (Supplementary Fig. 5d). Using the GEPIA2 web server to interrogate the Cancer Genome Atlas database[48], a combined CD69$^+$CD103$^+$ and γδ-TCR gene signature showed a favourable prognostic association with overall survival in HCC, not detected with an equivalent CD69$^+$CD103$^+$ αβCD8$^+$ TCR gene signature (Supplementary Fig. 5e), although the T$_{RM}$ gene signature in the analysis may have been contributed to by other cell types preventing definitive conclusions.

Overall, Vδ1 T-cells appeared higher in frequency and tended to be more activated with higher cytotoxic potential within HCC, while Vγ9Vδ2 T-cells, despite being selectively reduced in frequency in HCC, maintained high capacity for IFN-γ production and an equivalent ability to acquire an intratumoural CD69$^+$CD49a$^+$ or CD69$^+$CD103$^+$ T$_{RM}$ phenotype.

**Anti-tumour potential of Vγ9Vδ2 T$_{RM}$ against ZOL-sensitised HCC cell lines.** While both Vδ1 and Vγ9Vδ2 T-cells may contribute to immunosurveillance of HCC, we focused on therapeutic augmentation of the depleted Vγ9Vδ2 T-cell subset, for which ligands and clinical expansion protocols are better validated. Having demonstrated that Vγ9Vδ2 T$_{RM}$ were well-adapted for long-lived function in the HCC niche, we next investigated if they had the potential to mediate specific anti-tumour functionality. Intrahepatic Vγ9Vδ2 T-cells isolated from fresh human liver tissue exhibited minimal IFN-γ and TNF-α production in response to co-culture with HepG2 or HuH7 human hepatoma cell lines (Fig. 6a, b, Supplementary Fig. 6a). We hypothesised that insufficient isopentenyl pyrophosphate (IPP) production by the tumour cell lines may be limiting Vγ9Vδ2 TCR activation. Therefore, based on previous studies[51–53], we pre-treated HepG2 and HuH7 cells for 16–18 h with the aminobisphosphonate ZOL, which inhibits the enzyme farnesyl pyrophosphate synthase (FPPS) in the mevalonate pathway, promoting upstream

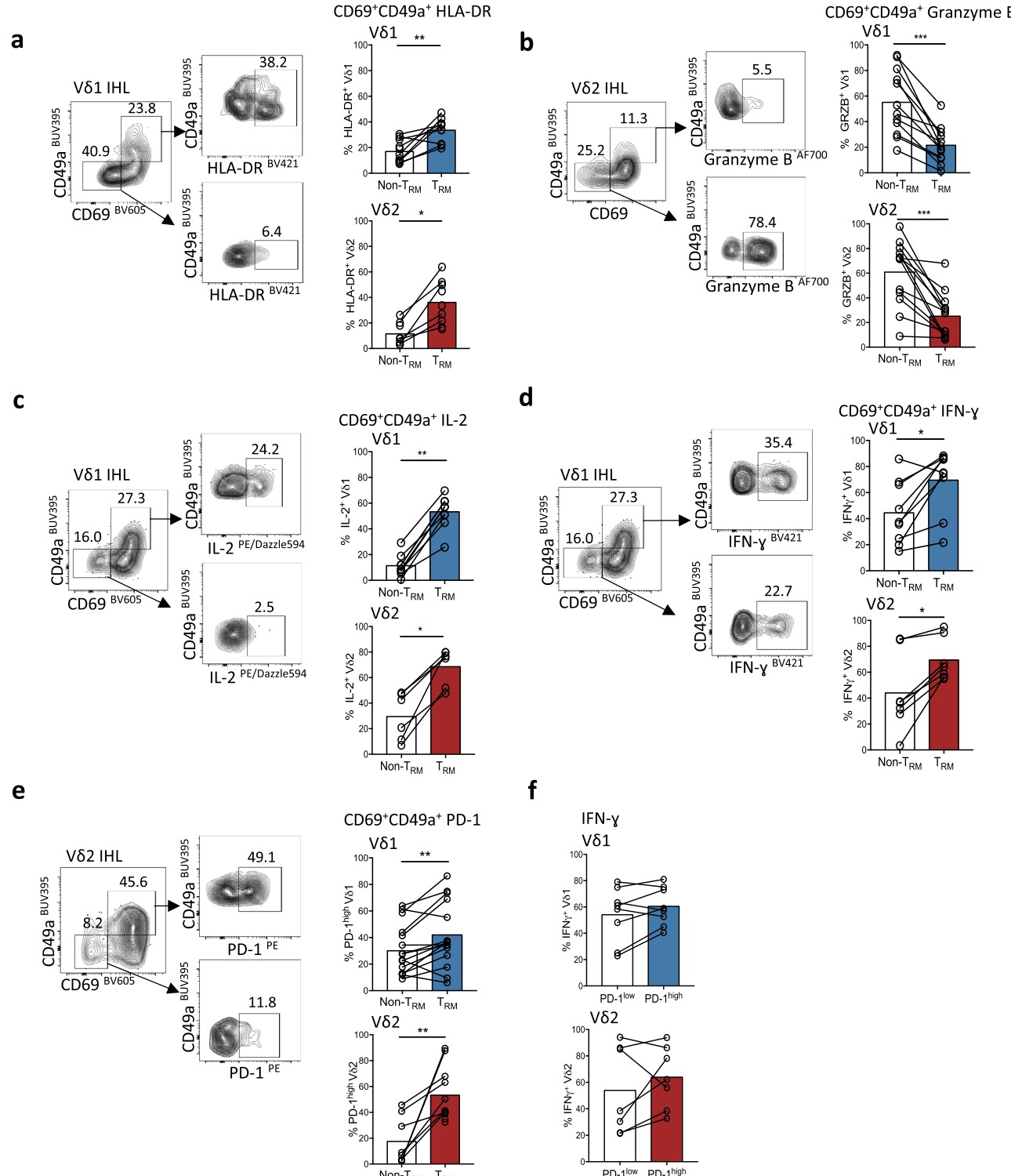

**Fig. 3 Distinct functional profile of hepatic γδT$_{RM}$. a–e** Representative flow cytometry plots and summary data of ex vivo functional profile of intrahepatic CD69$^+$CD49a$^+$ Vδ1 and Vγ9Vδ2 T$_{RM}$ compared to non-T$_{RM}$ (CD69$^-$CD49a$^-$) counterparts. a HLA-DR expression ($n$ = 13; Vδ1 $p$ = 0.008; Vγ9Vδ2 $p$ = 0.03). **b** unstimulated Granzyme B expression ($n$ = 17; Vδ1 $p$ = 0.0005; Vγ9Vδ2 $p$ = 0.0002). **c** IL-2 expression following 4 h PMA and Ionomycin stimulation ($n$ = 10; Vδ1 $p$ = 0.004; Vγ9Vδ2 $p$ = 0.03). **d** IFN-γ expression after 4 h PMA and Ionomycin stimulation ($n$ = 10; Vδ1 $p$ = 0.01; Vγ9Vδ2 $p$ = 0.02). **e** unstimulated programmed death-1 (PD-1) expression ($n$ = 11; Vδ1 $p$ = 0.007; Vγ9Vδ2 $p$ = 0.008). **f** IFN-γ expression by PD-1$^{high}$ and PD-1$^{low}$ intrahepatic Vδ1 and Vγ9Vδ2 T-cells after 4 h PMA and Ionomycin stimulation ($n$ = 8; $p$ value non-significant). Each symbol represents a study participant, with bars showing the mean (**a**–**f**); two-tailed $p$-values were determined using Wilcoxon matched-pairs signed rank test (**a**–**f**). *$p$ < 0.05; **$p$ < 0.01; ***$p$ < 0.001.

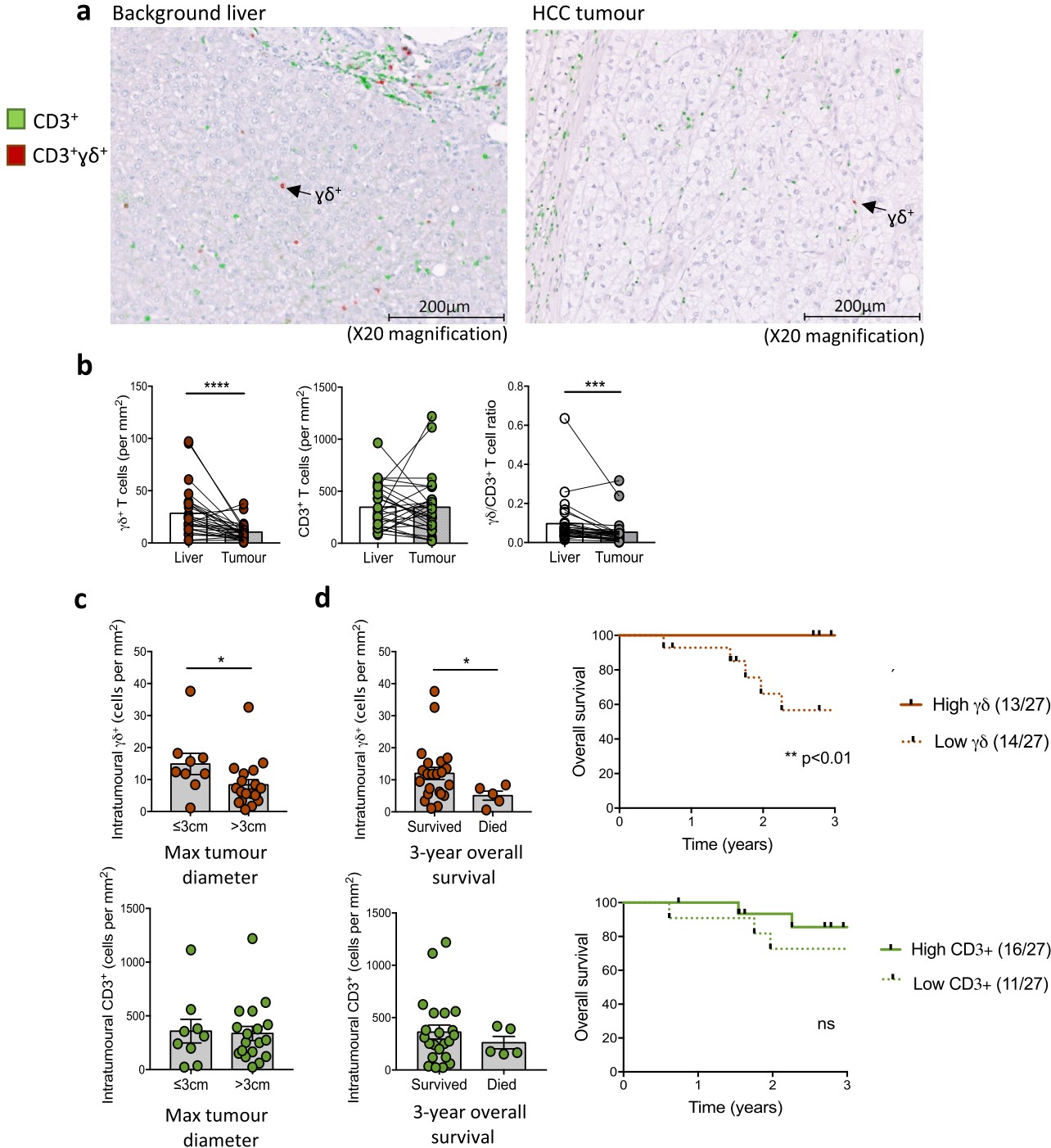

**Fig. 4 γδ T-cell counts in HCC are associated with tumour size and patient survival. (a–d)** Immunohistochemistry staining of paired background liver and tumour tissue obtained from patients with hepatocellular carcinoma (HCC) undergoing surgical resection. Cell counts performed from five randomly selected high-power fields (x20 objective magnification) per sample ($n = 28$ paired samples). **a** Representative multispectral analysis with non-γδ CD3+ (green) and γδ+ (red) T-cells as pseudo-colourised images in liver and HCC tumoural tissue, replicated across five high-power (x20) fields per sample. **b** Absolute γδ T-cell and non-γδ CD3+ T-cell numbers (calculated per mm$^2$) and γδ/CD3+ T-cell ratio in paired liver and HCC tumours ($n = 28$ paired samples; $p < 0.0001$, $p = 0.0006$). **c** Intratumoural γδ T-cell counts and non-γδ CD3+ T-cell counts (per mm$^2$) in small HCC tumours with a maximum diameter of ≤3 cm compared to HCC tumours >3 cm in diameter ($n = 28$; $p = 0.03$). **d** HCC intratumoural γδ T-cell counts and non-γδ CD3+ T-cell counts (per mm$^2$) according to 3-year patient survival outcomes (overall survival data available $n = 27$, 22/27 survived, 5/27 died; $p = 0.048$). Kaplan Meier graphs of overall survival (years post resection) split on the median intratumoural γδ T-cell ($p = 0.009$) or non-γδ CD3+ T-cell count from 27 HCC tumours. Two-tailed $p$-values were determined using Wilcoxon matched-pairs signed rank test test (**b**) or Mann–Whitney test (**c**, **d**), Kaplan Meier graphs with Log-rank test (**d**). Error bars represent mean ± SEM. ns Not significant; *$p < 0.05$; **$p < 0.01$; ***$p < 0.001$, ****$p < 0.0001$.

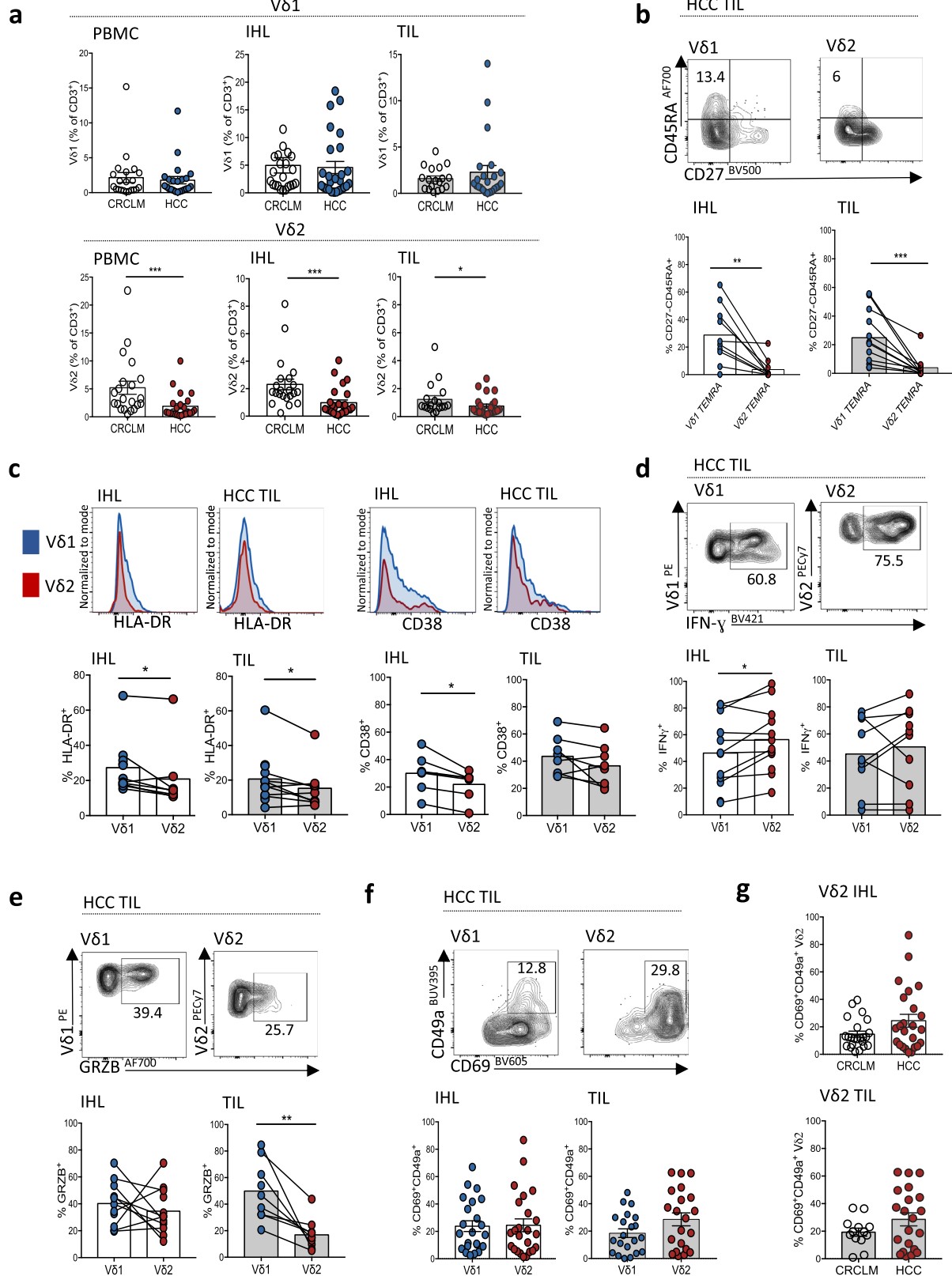

accumulation of IPP within the tumour cell to trigger Vγ9Vδ2 TCR activation[54]. After removal of ZOL, hepatoma cell lines were again co-cultured with IHL and were then able to trigger a significant increase in Vγ9Vδ2 T-cell effector function (IFN-γ, TNF-α, IL-2 expression, Fig. 6c, Supplementary Fig. 6a). This activation was specific to Vγ9Vδ2 T-cells, with no effect on liver Vδ1 or

αβCD8[+] T-cell function (Supplementary Fig. 6b). To confirm that the increase in Vγ9Vδ2 T-cell effector function was dependent on tumour cell IPP expression, we additionally treated ZOL-exposed hepatoma cell lines with Mevastatin, an inhibitor of the mevalonate pathway, able to block tumour cell IPP production, prior to the co-culture with IHL (Fig. 6d)[51,54]. Mevastatin

**Fig. 5 Vγ9Vδ2 T-cells are selectively depleted, but can acquire tissue-residence, within HCC TILs.** (**a–g**) Flow cytometry analysis of paired peripheral blood mononuclear cells (PBMC), intrahepatic lymphocytes (IHL) from tumour-free liver tissue, and tumour-infiltrating lymphocytes (TIL) from patients undergoing surgical resection or liver transplantation for hepatocellular carcinoma (HCC) compared to patients undergoing surgical resection for colorectal cancer liver metastases (CRCLM). **a** Total frequencies of Vδ1 (above) and Vγ9Vδ2 (below) T-cells in PBMC, IHL and TILs of CRCLM ($n = 23$) compared to HCC ($n = 26$) (expressed as a percentage of total CD3$^+$ T-cells; Vγ9Vδ2 PBMC $p = 0.0007$, IHL $p = 0.0002$, TIL $p = 0.02$). **b** Representative flow cytometry plot and summary data of CD27-CD45RA + terminally differentiated effector memory (TEMRA) Vδ1 and Vγ9Vδ2 T-cells in paired liver and HCC TILs ($n = 10$; IHL $p = 0.002$, TIL $p = 0.001$). **c** ex vivo HLA-DR ($n = 12$; IHL $p = 0.02$, TIL $p = 0.05$) and CD38 expression ($n = 8$; IHL $p = 0.02$) by Vδ1 and Vγ9Vδ2 T-cells within HCC IHL and TILs. **d** IFN-γ expression by Vδ1 and Vγ9Vδ2 T-cells within HCC IHL ($n = 12$) and TILs ($n = 10$) after 4 h PMA and Ionomycin stimulation (IHL $p = 0.03$). **e** Unstimulated Granzyme B expression by Vγ9Vδ2 T-cells in HCC IHL and TILs ($n = 11$; TIL $p = 0.008$). **f** CD69$^+$CD49a$^+$ expression on Vδ1 and Vγ9Vδ2 T-cells in HCC IHL ($n = 23$) and TILs ($n = 21$). **g** CD69$^+$CD49a$^+$ expression on Vγ9Vδ2 T-cells in HCC compared to CRCLM IHL ($n = 22$) and TILs ($n = 13$). Each symbol represents a study participant, error bars indicate mean ± SEM; two-tailed $p$-values were determined using Mann–Whitney test (**a, g**), or Wilcoxon paired test (**b–f**). *$p \leq 0.05$; **$p < 0.01$; ***$p < 0.001$.

significantly reduced the function of co-cultured intrahepatic Vγ9Vδ2 T-cells (IFN-γ and TNF-α), supporting a role for IPP-mediated triggering (Fig. 6d).

Importantly, CD69$^+$CD49a$^+$ Vγ9Vδ2 T$_{RM}$ isolated from the human liver displayed a greater increase in effector function (IFN-γ, IL-2 expression) compared to their non-resident (CD69$^-$CD49a$^-$) Vγ9Vδ2 T-cell counterparts when co-cultured with ZOL-treated hepatoma cell lines (Fig. 6e). We observed a similar response using TILs isolated directly from four HCC tumours. Co-culturing HCC TILs with plated HepG2 cells in vitro, resulted in no measurable Vγ9Vδ2 T-cell effector response (Fig. 6f). However, ZOL pre-treatment of the hepatoma cell lines increased IFN-γ expression by the intratumoural Vγ9Vδ2 T-cells, predominantly by the Vγ9Vδ2 T$_{RM}$ subset (Fig. 6f, g). Collectively, these results suggest that low IPP expression by HCC tumour cells could limit the local activation of Vγ9Vδ2 T-cells by HCC. Thus, by sensitising HCC tumour cells directly with ZOL first, we could significantly enhance the anti-tumour function of Vγ9Vδ2 T-cells, in particular Vγ9Vδ2 T$_{RM}$, within HCC tumours and adjacent liver tissue.

**ZOL expands de novo Vγ9Vδ2 T$_{RM}$ from blood, targeting ZOL-sensitised HCC cell lines.** Enhancing the function of residual endogenous Vγ9Vδ2 T$_{RM}$ is a promising novel immunotherapeutic strategy, however, its clinical efficacy may be limited by the low cell counts and reduced cytotoxicity of Vγ9Vδ2 T$_{RM}$ in HCC livers and tumours. To expand a large population of Vγ9Vδ2 T-cells suitable for immunotherapy, we used a well-established clinical protocol (see Methods), treating healthy donor PBMC with ZOL and IL-2 over 10 days[55–59]. Vγ9Vδ2 T-cells expanded with this protocol showed an unexpected de novo induction of a CD69$^+$CD49a$^+$ T$_{RM}$ phenotype (mean 80.3 ± 17.9%), with varying levels of CD103 co-expression (Fig. 7a, b). These de novo CD69$^+$CD49a$^+$ +/- CD103$^+$ Vγ9Vδ2 T$_{RM}$ displayed a CD27$^{low}$CD45RA$^{low}$ effector memory phenotype (Supplementary Fig. 7a), with downregulation of CD62L (Supplementary Fig. 7b). ZOL-induced Vγ9Vδ2 T$_{RM}$ recapitulated other features of Vγ9Vδ2 T$_{RM}$ analysed directly from the liver, with even higher levels of the liver-homing chemokine receptors CXCR6 and CXCR3 (Fig. 7c) and greater capacity for IFN-γ expression, although with lower PD-1 expression in comparison to ex vivo isolated intrahepatic Vγ9Vδ2 T$_{RM}$ (Fig. 7d, Supplementary Fig. 7c). Crucially, in vitro induced Vγ9Vδ2 T$_{RM}$ had enhanced cytotoxic potential (ex vivo granzyme B expression) compared to Vγ9Vδ2 T$_{RM}$ isolated directly from the liver (Fig. 7d).

ZOL inhibits FPPS within the mevalonate pathway of dendritic cells and monocytes in PBMCs, leading to accumulation of IPP to trigger Vγ9Vδ2 TCR activation[60–63], pointing to this as a putative mechanism for de novo Vγ9Vδ2 T$_{RM}$ induction. To further explore the role of TCR activation as a potential driving factor for

Vγ9Vδ2 T-cell tissue-residency, we stimulated Vγ9Vδ2 T-cells within PBMCs using a plate-bound anti-γδTCR antibody. Continuous TCR stimulation (7-days) induced Vγ9Vδ2 T-cell activation (increased CD38), accompanied by preferential de novo CD49a and CD103 expression on the CD38-expressing fraction (Supplementary Fig. 7d), accumulating in a time-dependent manner (Supplementary Fig. 7e), and accompanied by significantly higher levels of CXCR6 and CXCR3 expression (Supplementary Fig. 7f). IL-2 without TCR stimulation did induce a degree of CD49a expression on Vγ9Vδ2 T cells, although to a significantly lower intensity (Supplementary Fig. 7d, e). Exposure to other prototypic hepatic cytokines, in particular combined IL-15 and TGF-β, also induced de novo expression of CD49a, and to a much lower extent CD103, on Vγ9Vδ2 T-cells, and therefore may be additional factors driving the induction of γδ T$_{RM}$ within the liver and HCC (Supplementary Fig. 7g).

We next co-cultured the ZOL and IL-2 expanded blood Vγ9Vδ2 T-cells with human hepatoma cell lines, HepG2 and HuH7, to examine for an anti-HCC response. ZOL pre-treatment of the hepatoma cell lines was again required in order to evoke substantial IFN-γ and TNF-α expression by the expanded blood Vγ9Vδ2 T$_{RM}$ (Fig. 7e, Supplementary Fig. 7h), which was reduced in a dose-dependent manner by the addition of Mevastatin, confirming the dependence on tumour cell IPP expression (Fig. 7f, Supplementary Fig. 7i). Notably, ZOL-induced blood Vγ9Vδ2 T$_{RM}$ exhibited higher IFN-γ and Granzyme B expression than ex vivo isolated liver Vγ9Vδ2 T$_{RM}$ following co-culture with hepatoma cell lines (Fig. 7g). The addition of anti-PD-L1 blockade did not further increase expanded Vγ9Vδ2 T$_{RM}$ effector function towards ZOL pre-treated HepG2 cells (IFN-γ, TNF-α) (Fig. 7h). Critically, de novo induced Vγ9Vδ2 T$_{RM}$ were able to lyse HCC cell lines that had been sensitised by ZOL, assessed by the ToxiLight$^{TM}$ bioluminescent cytotoxicity assay measuring adenylate kinase in the culture medium released following cell death (Fig. 7i).

Overall, these data indicate that a de novo T$_{RM}$ phenotype with enhanced anti-tumour functionality can be successfully recapitulated and expanded in vitro from peripheral blood Vγ9Vδ2 T-cells, with liver-homing/retention potential for adoptive cell transfer. However, to enhance their anti-tumour function and maximise tumour cell lysis, prior treatment of HCC tumour cells with ZOL is required to upregulate IPP expression for Vγ9Vδ2 TCR activation in situ.

## Discussion
HCC is now known to be partially amenable to immunotherapy, but new approaches are urgently needed to tackle the dual tolerogenic influences of the liver and tumour microenvironments. T-cells specialised for tissue residence are adapted to combat local tolerising effects, and are able to maintain long-term, rapid

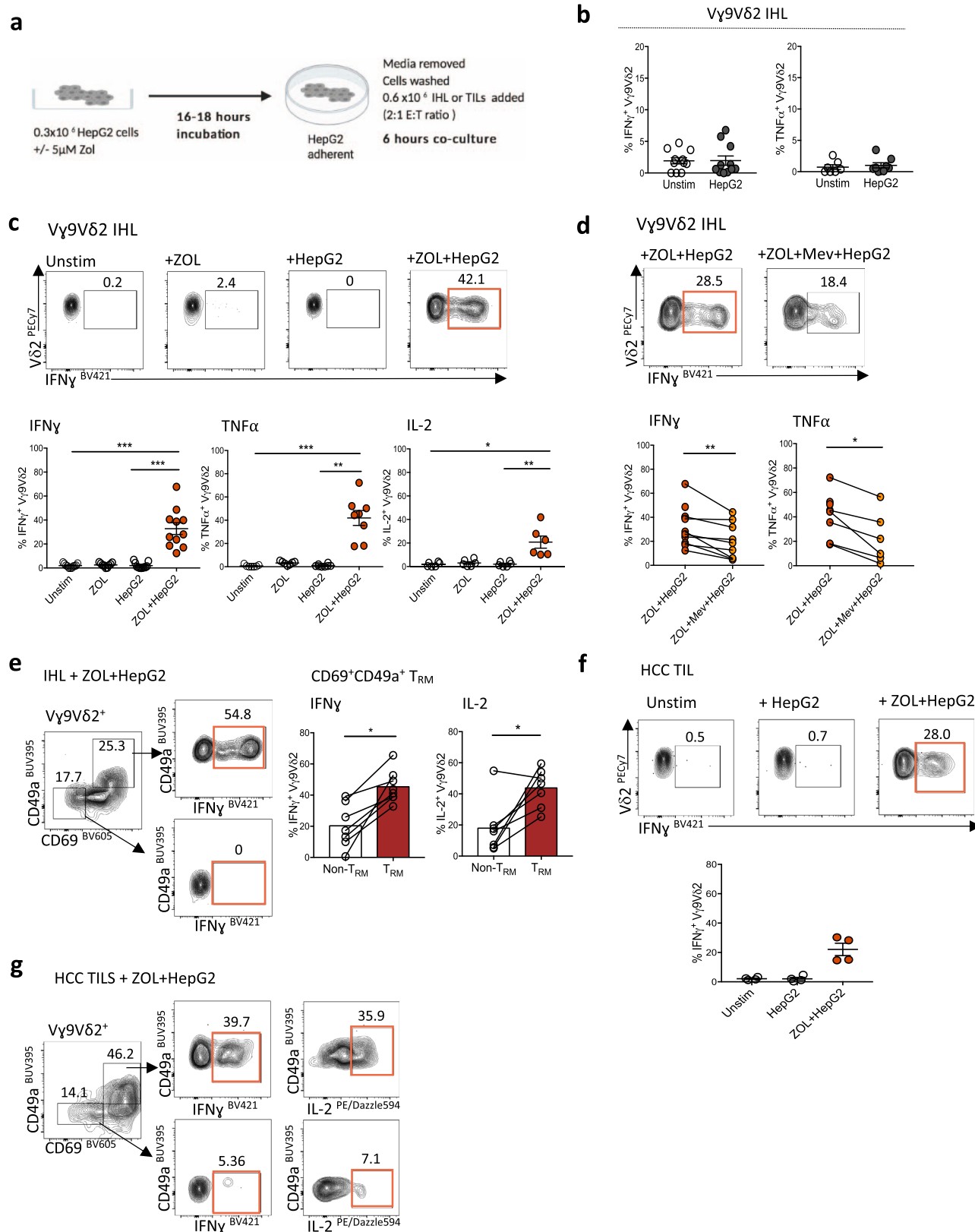

functionality at the site of disease. In the case of αβCD8+ T-cells, tissue-residence is emerging as a critical feature for protection against tumours[25,31–37,39]. γδ T-cells are promising alternative candidates for cancer immunotherapies but their capacity for tissue residency in HCC is unknown. Here, we show that both major subsets of human γδ T-cells (Vδ1 and Vγ9Vδ2) are able to acquire phenotypic and functional features of tissue residence in human HCC. We confirm the bona fide $T_{RM}$ status of CD69+CD49a+ γδ T-cells by demonstrating their lack of egress into hepatic vasculature, long-lived persistence in human liver allografts, and rapid production of cytokines favouring survival and non-cytolytic functionality, similar to intrahepatic αβCD8+

**Fig. 6 Anti-tumour potential of Vγ9Vδ2 T$_{RM}$ against ZOL-sensitised HCC cell lines. a** Schema of untreated or Zoledronic acid (ZOL) pre-treated (5μM, 16–18 h) adherent HepG2 cells co-cultured with intrahepatic lymphocytes (IHL) ($n = 11$) or HCC tumour-infiltrating lymphocytes (TIL) ($n = 4$) (E:T 2:1 ratio; $0.6 \times 10^6$ IHL or TILs to $0.3 \times 10^6$ HepG2 cells, 6 h co-culture, all conditions performed in duplicate or triplicate). **b** IFN-γ, TNFα expression by Vγ9Vδ2 T-cells in IHL unstimulated or after co-culture with untreated HepG2 cells ($n = 11$ IFN-γ, $n = 8$ TNF-α). **c** IFN-γ, TNFα, IL-2 expression by Vγ9Vδ2 T-cells in IHL: unstimulated, directly treated with 5 μM ZOL, or after co-culture with untreated or ZOL pre-treated HepG2 cells ($n = 11$ IFN-γ $p = 0.0004$, $p = 0.0008$; $n = 8$ TNFα $p = 0.0006$, $p = 0.004$; $n = 7$ IL-2 $p = 0.007$, $p = 0.02$). **d** IFN-γ ($n = 11$), TNF-α ($n = 8$) expression by Vγ9Vδ2 T-cells in IHL after co-culture with ZOL pre-treated HepG2 cells with or without 100 μM Mevastatin (Mev) treatment (IFN-γ $p = 0.008$, TNF-α $p = 0.03$). **e** Representative flow cytometry plot of IFNγ expression and summary data of IFNγ and IL-2 expression by CD69$^+$CD49a$^+$ Vγ9Vδ2 T$_{RM}$ compared to CD69$^-$CD49a$^-$ Vγ9Vδ2 non-T$_{RM}$, following co-culture of IHL with ZOL pre-treated HepG2 cells ($g = 8$, IFNγ $p = 0.02$, IL-2 $p = 0.03$). **f** IFN-γ expression by Vγ9Vδ2 T-cells in HCC TILs unstimulated, or after co-culture with untreated or ZOL pre-treated HepG2 cells ($n = 4$). **g** Representative flow cytometry plot of IFN-γ expression by CD69$^+$CD49a$^+$ Vγ9Vδ2 T$_{RM}$ and CD69$^-$CD49a$^-$ Vγ9Vδ2 non-T$_{RM}$ in HCC TILs after co-culture with ZOL pre-treated HepG2 cells. Each symbol represents a study participant; error bars represent the mean ± SEM. Two-tailed $p$-values were determined using Wilcoxon paired test (**b**, **d**, **e**), or Friedman test with Dunn's post-hoc multiple comparisons test (**c**, **f**). *$p < 0.05$; **$p < 0.01$; ***$p < 0.001$.

T$_{RM}$[28–30]. CD49a expression has similarly been described to promote αβCD8$^+$ T$_{RM}$ survival in the human lung, mediated in part through engagement with its ligand collagen IV[64].

The protective capacity of γδ T-cells against other cancer types has been demonstrated in mice and humans[9–16]; here, we show this also applies to HCC, with higher intratumoural γδ T-cell frequencies associating with smaller tumour size and longer patient survival. It will be important to extend this to much larger cohorts to assess for potential prognostic value as a clinical biomarker. Whilst the endogenous intratumoural γδ T-cell response may be partially limited by regulatory and/or pro-inflammatory influences[65–67], our findings reveal the potential to exploit the capacity of both Vδ1 and Vγ9Vδ2 T-cells to acquire tissue-residence within HCC. Here, we concentrated on exploring the immunotherapeutic potential of Vγ9Vδ2 T-cells since they were selectively depleted in HCC but can be readily expanded in vitro, have a well-defined tumour ligand in the form of IPP and showed the capacity to acquire T$_{RM}$ features. Although intratumoural Vγ9Vδ2 T-cells maintained efficient production of IFN-γ, a cytokine critical to the orchestration of tumour immunity[68], they had reduced cytotoxic potential, lower expression of T-cell activation markers, and poor recognition of HCC cell lines. Firstly, we showed that the anti-tumour function of existing endogenous intrahepatic and intratumoural Vγ9Vδ2 T$_{RM}$ could be enhanced in an IPP-dependent manner by pre-treating HCC cell lines with the aminobisphosphonate ZOL, as previously shown for circulating Vγ9Vδ2 T-cells[51]. However, this would not tackle the depleted numbers of Vγ9Vδ2 T-cells we observed in HCC. Next, we, therefore, explored the expansion of Vγ9Vδ2 T-cells with the combination and doses of ZOL and IL-2 previously optimised for adoptive cell transfer. We discovered that this clinically-validated protocol induced a de novo T$_{RM}$ phenotype on blood Vγ9Vδ2 T cells, with increased liver-homing chemokine receptor expression and even higher expression of the retention molecules CD69/CD49a/CD103 than endogenous γδT$_{RM}$, as well as enhancing cytotoxicity. Analogous to the endogenous Vγ9Vδ2 T-cell response, these peripheral induced Vγ9Vδ2 T$_{RM}$ were able to efficiently recognise and lyse HCC cells pre-sensitised with aminobisphosphonates to increase IPP accumulation.

Thus, Vγ9Vδ2 T-cell-based immunotherapy holds potential as an alternative treatment strategy for HCC patients resistant or intolerant to immune checkpoint inhibition, removing the need for tumour-antigen specificity due to the uniform, MHC-unrestricted responsiveness of Vγ9Vδ2 T-cells to phospho-antigen (IPP) stimulation. The adoptive cell transfer of ex vivo ZOL-expanded blood Vγ9Vδ2 T-cells has demonstrated a good safety profile in several early phase clinical trials[55–59], including a recent phase one trial in patients with advanced HCC[69]. However, to date, clinical efficacy has remained low. Systemic ZOL therapy can expand Vγ9Vδ2 T-cells in vivo, but the high affinity of ZOL for bone mineral and its rapid renal clearance limits

systemic availability of ZOL for tumour cell uptake[70], likely contributing to the insufficient anti-tumour responses observed[22,71–73]. In contrast, local intraperitoneal or intratumoural administration of ZOL to tumours has demonstrated early promise in small *proof-of-concept* studies of other tumour types[52,53,57,74]. Our data suggest that aminobisphosphonates should be tested in a combination strategy for HCC, to overcome two key limiting factors: their direct delivery to the tumour can enhance IPP expression to increase in situ activation of Vγ9Vδ2 T-cells, whilst their use for large-scale expansion of blood Vγ9Vδ2 T-cells can expand a population with enhanced cyto-toxicity and the tissue-retention properties of T$_{RM}$ to replenish depleted numbers within the tumour.

## Methods

**Study approval.** This study was approved by the local ethics board Brighton and Sussex (Research Ethics Committee reference number 11/LO/0421) and complies with the Declaration of Helsinki. All study participants provided written informed consent before inclusion. All storage of samples obtained complied with the requirement of the Human Tissue Act 2004 and the Data Protection Act 1998.

**Sample collection.** All uses of human samples have been approved. Surgically resected tumour-free liver tissue and tumours of HCC or CRCLM, explants from patients with HCC or cirrhosis without HCC undergoing liver transplantation, and perfusates from healthy donor liver allografts, together with paired blood samples, were obtained through the Tissue Access for Patient Benefit initiative at The Royal Free Hospital (approved by the University College London–Royal Free Hospital BioBank Ethical Review Committee; Research Ethics Committee reference number 11/WA/0077). Formalin fixed paraffin-embedded tissue sections for immunohistochemistry were obtained from background liver and tumours of HCC (approval by the Royal Free Hospital Ethics Committee reference number 07/Q0501/50). Blood samples were obtained from patients with cirrhosis undergoing transjugular intrahepatic portosystemic shunt procedures (approved by the University College London–Royal Free Hospital BioBank Ethical Review Committee; Research Ethics Committee reference number 16/WA/0289). For comparison, peripheral blood samples from healthy control individuals were included within the study (approved by the South East Coast Research Ethics Committee; Research Ethics Committee reference number 11/LO/0421).

Explanted liver tissue was obtained from two patients undergoing re-transplantation where there was an HLA class I mismatch between the initial allograft donor and the recipient (approved by the University College London–Royal Free Hospital BioBank Ethical Review Committee; Research Ethics Committee reference number 11/WA/0077). HLA phenotype was determined by HLA-haplotyping PCR by A. Nolan (National Health Service, London, UK) or MRC Weatherall Institute of Molecular Medicine Sequencing Facility (Oxford, UK). Clinical details of the transplant recipients and donors are included in Supplementary Table 1.

**Human PBMC, IHL and TIL cell isolation.** Peripheral blood mononuclear cells (PBMC) were isolated from heparinised blood by density centrifugation using Pancoll (Pan Biotech) and was centrifuged for 20 min at 800 g with slow acceleration and deceleration. PBMCs were used fresh where possible or were frozen in 10% DMSO (Sigma-Aldrich) in heat-inactivated FBS (Sigma-Aldrich) and stored in accordance with the Human Tissue Act.

To isolate intrahepatic lymphocytes (IHL) and tumour-infiltrating lymphocytes (TIL), liver or tumour tissue was cut into small pieces and incubated for 30 min at 37 °C in 0.01% collagenase IV (Thermo Fisher Scientific) and 0.001% DNAse I

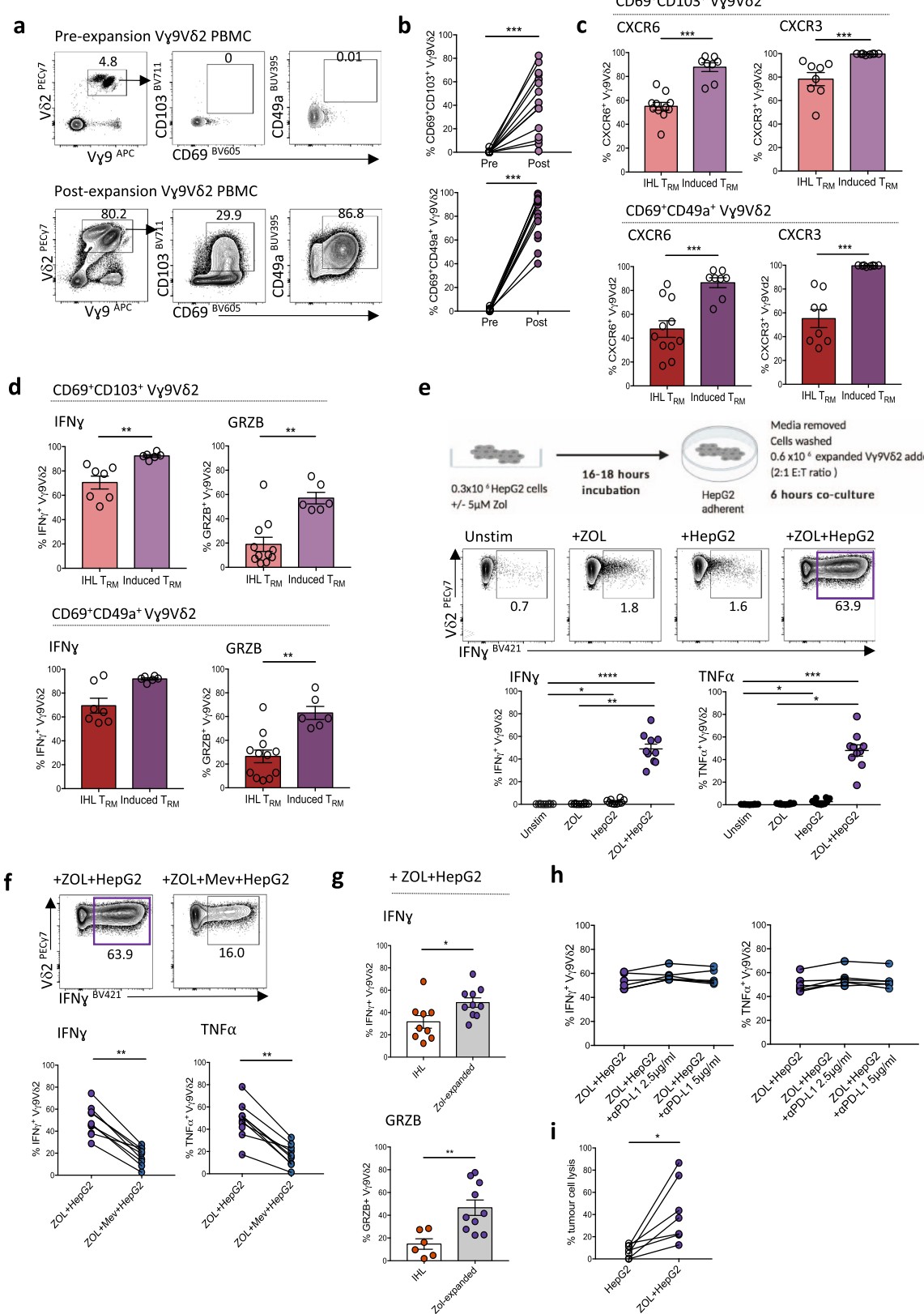

(Sigma-Adrich) followed by further mechanical disruption via GentleMACS (Miltenyi Biotech), and filtration through a 70 μm cell strainer (BD Bioscience) to achieve a single cell solution. Parenchymal cells were removed by 400 g centrifugation on a 30% Percoll gradient (GE Healthcare) and lymphocytes were isolated via density centrifugation as described above. IHL and TILs were used immediately after isolation.

**Flow cytometry: cell surface, intracellular and cytokine staining.** Multi-parametric flow cytometry was used for phenotypic and functional analysis of PBMC, IHL and TILs. Cells were stained with a fixable Live/Dead dye (Invitrogen) before incubation with saturating concentrations of surface mAbs diluted in 50% Brilliant Stain buffer (BD Biosciences) and 50% PBS (Invitrogen) for 15 min at 4 °C. Cells were fixed and permeabilized for further functional assessment with

**Fig. 7 ZOL expands de novo Vγ9Vδ2 T$_{RM}$ from blood, targeting ZOL-sensitised HCC cell lines. a** Representative flow cytometry plots of CD69$^+$CD103$^+$ and CD69$^+$CD49a$^+$ expression on Vγ9Vδ2 T-cells pre- and post-Zoledronic acid (ZOL) and IL-2 based expansion. **b** Summary data of CD69$^+$CD103$^+$ and CD69$^+$CD49a$^+$ expression on Vγ9Vδ2 T-cells pre- and post-ZOL and IL-2 based expansion ($n = 15$, $p = 0.001$). **c** % CXCR6 (left) and % CXCR3 (right) expression on ex vivo intrahepatic Vγ9Vδ2 T$_{RM}$ ($n = 11$), compared to de novo blood CD69$^+$CD103$^+$ or CD69$^+$CD49a$^+$ Vγ9Vδ2 T$_{RM}$ (induced T$_{RM}$, $n = 8$) following ZOL/IL-2 expansion (CXCR6 $p = 0.0001$, $p = 0.0008$; CXCR3 $p = 0.0002$, $p = 0.0002$). **d** IFN-γ expression (left) by ex vivo intrahepatic Vγ9Vδ2 T$_{RM}$ compared to induced blood Vγ9Vδ2 T$_{RM}$ after 4 h PMA and Ionomycin stimulation ($n = 7$, $p = 0.001$); unstimulated Granzyme B expression (right) by intrahepatic Vγ9Vδ2 T$_{RM}$ ($n = 11$) compared to induced blood Vγ9Vδ2 T$_{RM}$ ($n = 6$, $p = 0.002$, $p = 0.001$). **e** Schema demonstrating expanded blood Vγ9Vδ2 T-cells co-culture with untreated or ZOL pre-treated (5 μM, 16–18 h) HepG2 cells (E:T 2:1 ratio; 0.6 × 10$^6$ expanded Vγ9Vδ2 T-cells to 0.3 × 10$^6$ HepG2 cells, 6 h co-culture, all conditions performed in triplicate). Representative flow cytometry plot and summary data of IFN-γ and TNF-α expression by ZOL/IL-2 expanded Vγ9Vδ2 T-cells: unstimulated, directly treated with ZOL 5 μM, or after co-culture with untreated or ZOL pre-treated HepG2 cells ($n = 10$; IFN-γ $p < 0.0001$, $p = 0.03$, $p = 0.01$; TNF-α $p < 0.001$, $p = 0.03$, $p = 0.01$). **f** IFN-γ and TNF-α expression by ZOL/IL-2 expanded Vγ9Vδ2 T-cells after co-culture with ZOL pre-treated HepG2 cells, with or without the addition of 100 μM Mevastatin (Mev) ($n = 10$; IFN-γ $p = 0.004$, TNF-α $p = 0.004$). **g** IFN-γ and Granzyme B expression by expanded blood Vγ9Vδ2 T$_{RM}$ ($n = 9$) compared to ex vivo intrahepatic Vγ9Vδ2 T$_{RM}$ ($n = 10$) after co-culture with ZOL pre-treated HepG2 cells (IFN-γ $p = 0.03$, GRZB $p = 0.005$). **h** IFN-γ and TNF-α expression by ZOL/IL-2 expanded Vγ9Vδ2 T-cells after co-culture with ZOL pre-treated HepG2 cells, with or without the addition of anti-programmed death-ligand 1 (PDL-1) blockade ($n = 6$). **i** Lysis of HepG2 cells and ZOL pre-treated HepG2 cells after co-culture with PBMCs containing ZOL-expanded Vγ9Vδ2 T-cells ($n = 7$, $p = 0.02$), measured using Toxilight$^{TM}$ cytotoxicity assay. Each symbol represents a study participant; error bars show mean ± SEM. Two-tailed $p$-values determined using Wilcoxon paired test (**b**, **f**, **i**), Mann–Whitney test (**c**, **d**, **g**), or Friedman test with Dunn's post-hoc test for multiple comparisons (**e**, **h**). \*$p < 0.05$; \*\*$p < 0.01$; \*\*\*$p < 0.001$; \*\*\*\*$p < 0.0001$.

---

Cytofix/Cytoperm (BD Biosciences) according to the manufacturer's instructions. Cells were incubated with saturated concentrations of mAbs diluted in a 0.1% saponin-based buffer (Sigma-Aldrich) for 20 min at 4 °C for the detection of intracellular proteins. All samples were acquired on Fortessa X20 flow cytometer using FACSDiva software v8.0 (BD biosciences) and analysed using FlowJo v.9 (Tree Star; BD Biosciences). Full details on monoclonal antibodies used can be found in Supplementary Table 5.

**t-Distributed Stochastic Neighbour Embedding (tSNE) analysis**. The dimension reduction algorithm tSNE was applied to concatenated flow cytometry data (~2500 cells per samples) from 5 IHL and 3 PBMC samples using default parameters (iterations, 1,000; perplexity, 30; and eta 4076, vantage point tree KNN algorithm) in FlowJo. tSNE was applied to expression data for Vδ1, Vδ2, Vγ9, CD69, CD49a, CD103, CXCR3, CXCR6, HLA-DR, NKG2D, after pre-gating on single, live, CD45 + , CD3 + , γδ-TCR + lymphocytes. Manual gating of Vδ1, Vδ2, Vγ9, CD69, CD103, and CD49a were used to colour subpopulations within the plots in Fig. 1b.

**In vitro stimulation**. Prior to intracellular cytokine staining, PBMC, IHL or TILs were plated in a 96-well plate (0.4 x 10$^6$ cells per well) in complete RPMI (cRPMI; RPMI-1640 containing 10% FBS, 100 U/ml penicillin/streptomycin, 1× non-essential amino acids, 1× essential amino acids, and β-mercaptoethanol; Invitrogen) and stimulated with 50 ng/ml phorbol myristate acetate (PMA) (Sigma-Aldrich) and 500 ng/ml Ionomycin (Sigma-Aldrich) for 4 h with 1 μg/ml brefeldin A (Sigma-Aldrich) added, at 37 °C in a humidified atmosphere with 5% CO$_2$. After stimulation, γδ T-cells were stained for cytokine production as above and analysed by flow cytometry.

For assessment of the impact of anti-PD-L1 receptor blockade, IHL were plated in 200 μl cRPMI in a 96-well plate (0.4 x 10$^6$ cells/well) with or without anti-PD-L1 blocking antibody (Invitrogen, 2.5 μg/ml or 5 μg/ml) and plate-bound anti γδ-TCR monoclonal antibody stimulation (Biolegend, 4 μg/ml), with 1 μg/ml brefeldin A (Sigma-Aldrich) added. After 16 h, cells were removed from stimulation, stained for cytokine production and analysed by flow cytometry.

**Immunohistochemistry staining of CD3$^+$ and γδ T-cells**. The immunostaining protocol was first optimised on formalin-fixed paraffin-embedded tonsil tissue as a positive control, and subsequently liver tissue. Formalin-fixed paraffin-embedded sections obtained from paired HCC and background liver tissue ($n = 28$) were placed through xylene and alcohol to distilled water, placed in 1 L of pH 9.0 Tris EDTA buffer and then microwaved for 20 min at 640 W before cooling and rinsing in running tap water. The slides were soaked in Tris Buffered Saline (TBS) with 0.04% Tween-20 for 5 min, Bloxall (Vector – SP-6000-100) blocking solution was applied for 10 min, washed in TBS and blocked with normal horse serum (MP-7402 – Vector Laboratories) for 10 min. Slides were incubated for 1 h at room temperature in a combined cocktail of rabbit anti human CD3 (SP7 clone, ab16669-Abcam) at 1:100 and mouse anti human TCR-δ (H41 clone, sc-100289-Santacruz) at 1:100 in TBS. After washing in TBS Tween for 5 min, the slides were incubated for 25 min in ImPRESS horse anti-mouse HRP reagent (MP-7402-Vector Laboratories), washed in TBS Tween, developed with ImmPACT DAB (SK4105-Vector Laboratories), washed in TBS, and blocked with normal horse serum (MP5401–Vector Laboratories) for 10 min. Slides were incubated for 25 min in ImPRESS horse anti-rabbit HRP reagent (MP5401-Vector Laboratories), washed in TBS Tween, developed with fast red alkaline phosphatase substrate

(ab64254=Abcam), washed in TBS Tween, counterstained with Mayers haematoxylin and then rinsed in running tap water. Slides were allowed to air-dry to dehydrate, clear in xylene and mounted. All relevant negative controls and single stains were included to exclude the possibility of cross reaction between CD3 and TCR-δ antibodies and detection kits.

**Immunohistochemistry imaging and quantification of CD3$^+$ and γδ T-cells**. Five fields were selected at random from each of the background liver ($n = 28$) and HCC tumour ($n = 28$) slides using a graticule and a random number generator, and enlarged to x20 objective magnification. Images were taken using a Akoya Mantra 2$^{TM}$ microscope and multispectral camera which captures an absorbance spectrum for each pixel. Spectral libraries were set up from single stained slides for each of DAB, Fast Red and Haematoxylin to identify the chromogen for TCR-γδ, CD3 and nuclei, respectively. The images were then spectrally unmixed using the spectral libraries to identify each chromogenic signal. Cells were identified using automatic segmentation with the software available from Akoya, inForm®, using the multispectral data to identify cells based on morphological parameters and chromogen expression. Automatic cell counts of CD3$^+$ positive cells and CD3$^+$TCR-γδ$^+$ cells were carried out on each image and segmented images were reviewed for accuracy of detection. Overall CD3$^+$ and CD3$^+$ TCR-γδ$^+$ cell counts and a count per mm$^2$ were obtained.

**Expansion of Vγ9Vδ2 T-cells**. PBMC from healthy controls were plated in a 24-well plate (0.2 × 10$^6$ cells/well) and incubated with 1000 IU/ml IL-2 (PeproTech) in the presence or absence of 5 μM Zometa (Novartis), in cRPMI at 37 °C for 3-days. On day 3 the PBMCs were removed, washed in cRPMI and re-plated in a 24-well plate (0.2 × 10$^6$ cells/well) with fresh cRPMI media containing 1000IU/ml IL-2. Every 2–3 days a media change was performed with 1000 IU/ml IL-2 added and the cells were split as required. On day 9–12, the cells were stained for Vγ9Vδ2 T-cell frequency and phenotype and analysed by flow cytometry. CountBright$^{TM}$ absolute counting beads (ThermoFisher) were used to determine total cell counts to confirm Vγ9Vδ2 T-cell expansion.

For comparisons of cytokine production post-expansion, following the Vγ9Vδ2 T-cell expansion protocol as described, PBMC were rested overnight in cRPMI and then plated in a 96-well plate (0.4 x 10$^6$ cells/well) and stimulated using 50 ng/ml PMA (Sigma-Aldrich) and 500 ng/ml Ionomycin (Sigma-Aldrich) for 4 h at 37 °C with 1 μg/ml brefeldin A (Sigma-Aldrich) added. After stimulation, cells were stained for cytokine production and analysed by flow cytometry.

**In vitro γδ-TCR activation**. PBMC from healthy controls were plated in 200 μl cRPMI in a 96-well plate (0.3 x 10$^6$ cells/well) with IL-2 (20 IU/ml) added with or without additional plate-bound anti-γδTCR monoclonal antibody stimulation (Biolegend, 4μg/ml). PBMC were removed at 4 h, 24 h, 3 days, and 7 days of culture and stained for CD38, CXCR6, CXCR3 and γδ T$_{RM}$ marker expression.

**In vitro cytokine exposure**. PBMC from healthy controls were plated in a 96-well plate (0.3 x 10$^6$ cells/well) and cultured in 200 μl cRPMI with combinations of IL-2 (20 IU/ml), IL-7 (50 ng/ml), IL-15 (50 ng/ml), TGF-β (50 ng/ml). On day 7 PBMC were stained for cell surface γδ T$_{RM}$ marker (CD69, CD103, CD49a) expression.

**Zoledronic acid treated hepatoma cell line co-culture**. HepG2 or HuH7 cells were plated at a density of 0.3 x 10$^6$ cells/well in a 24-well plate (Costar) and

incubated in GlutaMax DMEM media (Gibco) containing 10% FCS, 1% NEAA, 1% Sodium Pyruvate, 1% PenStrep, for 16–18 h to adhere to the well, in the presence or absence of Zometa (ZOL) 5 μM (Novartis) and/or Mevastatin 100 μM or 200 μM (Sigma-Aldrich) and/or 5 μg anti-PD-L1 blocking antibody (Invitrogen). After 16–18 h incubation, the media was carefully removed and the adherent HepG2 cells in each well were washed twice with PBS. PBMC containing ZOL-expanded blood Vγ9Vδ2 T-cells, bulk IHL or TILs (0.6 x 10⁶ cells/well, E:T ratio 2:1) were then added to each well in cRPMI for 6 h at 37 °C with 1 μg/ml brefeldin A added in culture. All conditions were performed in duplicate or triplicate and combined before staining. Cells were removed, stained for Vγ9Vδ2 T-cell phenotype and effector function and analysed by flow cytometry.

**HCC cell line cytotoxicity assay.** HepG2 cells were plated (0.3 x 10⁶ cells/well) in a 24-well plate (Costar) and incubated in GlutaMax DMEM media containing 10% FCS, 1% NEAA, 1% Sodium Pyruvate, 1% PenStrep for 16–18 h, in the presence or absence of Zometa (ZOL) 5 μM (Novartis). After incubation, adherent HepG2 cells were washed twice with PBS. ZOL treated PBMCs (containing expanded Vγ9Vδ2 T-cells) were subsequently added to each well in cRPMI for 6 h at 37 °C. The culture supernatant was collected, frozen at −20 °C and thawed at time of analysis. Control wells included ZOL Vγ9Vδ2-expanded PBMCs alone, HepG2 cells alone and ZOL pre-treated HepG2 cells alone. 100% lysis positive controls included HepG2 cells and ZOL pre-treated HepG2 cells resuspended in 0.1% Triton X-100 in PBS. ToxiLight^TM bioluminescent cytotoxicity assay (Lonza) was used to measure adenylate kinase (AK) release in the culture supernatant as a marker of cell death. As per manufacturer's instructions, samples were mixed with the reconstituted AK detection agent, incubated for 5 min and then placed in a luminescence compatible plate reader. Specific lysis was calculated as follows, referring to ZOL Vγ9Vδ2-expanded PBMCs as effector cells and HepG2 cells as target cells: (RLU): (RLU(Effector+Target)-RLU(Effector)-RLU(Target))/(RLU(100%lysis)-RLU(Effector)-RLU(Target))*100.

**Statistical analyses.** Statistical analyses were performed in Prism 7.0 (GraphPad) using appropriate methods as indicated in the figure legends (Mann–Whitney test, Wilcoxon matched-pairs signed rank test, Spearman rank correlation test, Kruskal–Wallis test for unpaired non-parametric multiple comparisons, Friedman test for paired non-parametric multiple comparisons, with Dunn's post-hoc test for multiple comparisons) with significant differences marked on all figures. All tests were performed as two-tailed tests, and for all tests significance levels were defined as: not significant (ns) $p > 0.05$; *$p < 0.05$; **$p < 0.01$; ***$p < 0.001$; ****$p < 0.0001$.

**Reporting summary.** Further information on research design is available in the Nature Research Reporting Summary linked to this article.

## Data availability
The raw data for Figs. 1–7 and Supplementary Figs. 1–7 are provided in the Source Data file. The gene expression analysis data obtained from the Cancer Genome Atlas database are publicly available through the Gene Expression Profiling Interactive Analysis 2 (GEPIA2) web server, http://gepia2.cancer-pku.cn/#survival[48]. Source data are provided with this paper.

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

## Acknowledgements

This work was funded by a Wellcome Trust PhD Clinical Research Fellowship Award (175479) to N.Z., and by a CRUK Accelerator award HUNTER, CRUK Immunology project grant (26603) and Wellcome Trust Investigator Award (214191/Z/18/Z) to M.K.M. A.Q. is supported by the National Institute for Health Research (NIHR) UCLH/UCL Biomedical Research Centre (BRC). We are grateful to patients participating in the study and to clinical staff who helped with participant recruitment and sample acquisition, including NHS transplant coordinators, research nurses and the Tissue Access for Patient Benefit (TapB) project at The Royal Free Hospital (RFH) funded by RFH Charity and UCLH/UCL BRC, as well as Jamie Evans in the Rayne Building Flow Cytometry Core Facility who provided Fortessa X20 support. The illustrations in Figs. 2a, c, 6a, 7e were created with BioRender.com, with a granted publication license.

## Author contributions

N.Z. and M.K.M. conceived the project; N.Z., A.H., M.K.M. designed experiments; N.Z. and A.H. generated data; N.Z., A.H., A.Q., M.K.M. analysed and interpreted data; N.Z., A.H., L.S., L.J.P., N.S., M.D., S.K., O.A., A.G., M.P., B.D., A.Q., M.K.M. provided or processed essential patient samples and clinical data. N.Z. and M.K.M. prepared the manuscript. All authors provided critical review of the manuscript.

## Competing interests

Unrelated to the content of this manuscript, authors M.K.M. and N.M.S. have an international patent application No.1917498.6 entitled Treatment of Hepatitis B Virus (HBV) Infection filed by applicant UCL Business Ltd. MP is co-founder and director of Engitix Therapeutics Ltd, UK. The Maini lab has received unrestricted funding from Gilead, Roche and Immunocore. The remaining authors declare no competing interests.
