## [Peer Review File · Nature Communications]

Characterisation and induction of tissue-resident gamma delta T-cells to target hepatocellular carcinomaREVIEWER COMMENTS

Reviewer #1 (Remarks to the Author):

The Manuscript by Zakery N investigate the role of tissue resident gamma delta T cells in hepatocellular carcinoma, highlighting their possible use in cancer immunotherapy. The paper is quite nice, well written and correctly referenced. It reports experiments that are well designed, data which are convincing, and well interpreted. Some very minor revisions are nevertheless required about text and typo errors, which nevertheless do not affect the important significance of this study.

Reviewer #2 (Remarks to the Author):

The authors showed that the intra-tumoral $\gamma\delta$ T cells play an important role in cancer immunosurveillance of HCC patients. They characterized the tissue-resident features of $\gamma\delta$ T cells even in a circulating phenotype $V\gamma9V\delta2$ T cells by showing their lack of egress into hepatic vasculature and long-lived persistence in human liver transplanted allografts. The number of $V\gamma9V\delta2$ T cells was decreased in the IHL, TIL and PBMC in HCC patients, they acquired tissue-residence, maintaining the ability of producing anti-tumor cytokines and exerted cytotoxicity against the cancer cells pre-treated with zoledronic acid (ZOL). Finally, in vitro induced $V\gamma9V\delta2$ T cells recapitulated a resident memory phenotype and kill ZOL-treated cancer cells.

This manuscript is well written and very interesting in that the authors fully characterized the tissue-resident features of $\gamma\delta$ T cells especially in $V\gamma9V\delta2$ T cells in detail and gave a rationale for expanded $V\gamma9V\delta2$ T cell transfer therapy with local injection of aminobisphosphonate in HCC patients, although this therapeutic strategy may be the same as the approaches previously reported. Thus, my questions and comments are:

1. In IHL, CD69+CD49a+ T cells included central memory (CD27+CD45RA-) and effector memory (CD27-CD45RA-) T cells (Supplementary Fig.1d). Did the author check the HCC TIL? For example, more effector memory or terminally differentiated $\gamma\delta$ T cells were detected?
2. In Fig. 4, $\gamma\delta$ T-cell counts in the tumor was correlated with tumor size and patient prognosis. Which subset $V\delta1$ or $V\delta2$ T cells contribute to immunosurveillance dominantly? Though $V\gamma9V\delta2$ T cells were enriched for expression of resident markers within HCC TILs, the frequency of $V\gamma9V\delta2$ T cells was lower than that of $V\delta1$ T cells in the liver and tumor.
3. For the strategy of intra-tumoral delivery of zoledronate combined with expanded $V\gamma9V\delta2$ T cell therapy, systemic injection (Kobayashi et al, Cancer Immunol Immunother, 2011) or local (Intraperitoneal) injection (Wada et al, Cancer Med, 2014; ref. 52 in this manuscript) of zoledronate before $V\gamma9V\delta2$ T cell infusion had been tried previously. But their efficacies seem to be limited. What do the authors think these results? I recommend the authors discuss this.
4. In Fig. 3e, it was shown that IHL CD69+CD49a+ $\gamma\delta$ T cells expressed high PD-1. Do the authors think that the possibility of further combination of immune check point inhibitors with ZOL-based $\gamma\delta$ T cell therapy?

Reviewer #3 (Remarks to the Author):

Zakeri and colleagues investigate an interesting topic, namely tissue-resident $\gamma\delta$ T cells in the human liver. The authors characterize tissue-resident (Trm) $\gamma\delta$ T cells with a CD49a+, CD103+ and CD69+ phenotype, and these comprise $V\delta1+$ as well as $V\delta2+$ $\gamma\delta$ T cells. Their study involves the examination of $\gamma\delta$ T cells isolated from two explant samples in which donor and recipient $\gamma\delta$ T cells could be distinguished via HLA haplotypes. Donor-derived $\gamma\delta$ T cells showed a Trm phenotype.

Further, the authors suggest that higher counts of $\gamma\delta$ T cells in HCC are correlated to better outcome/ survival. They propose that (ex vivo) activation and transfer or in vivo activation of phosphoantigen-reactive $V\gamma9V\delta2$ cells would be a suitable therapy to target HCC.

Overall, I am not convinced that this manuscript combines its several findings to one coherent story.

Major issues:

It would be helpful to focus on the Trm gd T cells in the liver and e.g. extend the Trm signature beyond and or confirm their signature by transcriptional data.

If the authors want to make the point that activation of Vg9Vd2 cells leads to liver residency, this should be shown, at least by analysis of their liver homing capacity

Minor issues:

- Figure 1. Compartmentalization of Vd1 and Vγ9Vd2 T-cells with a tissue-resident phenotype in human liver. Are the different levels of expression of CD49a and CD103 associated with age / sex or any other clinical parameter of patients?
- Fig 1b: it seems like Vd1+ and Vd1- are not separated
- The pan gd TCR staining in Fig 2c,d is not convincing, please show full gating strategies or at least all T cells or all lymphocytes in a gdTCR vs. HLA-A2 plot.
- Figure 2. Long-lived hepatic retention and replenishment of Vd1 and Vγ9Vd2-TRM: fig 2a analysis limited to 3 patients, hard to draw conclusion considering variability among patients. Fig 2d should show data on gated vd1 and Vd2 rather than pan-gd. Since it's only two patients the data could be summarized with dimensional reduction analysis as U-MAP or t-SNE.
- Figure 3. Distinct functional profile of hepatic gdTRM. Fig. 3e plot showing PD1 expression among CD69+CD49a+ Vd2 cell should be representative of statistical mean.
- Figure 4. gd T-cell counts in HCC are associated with tumour size and patient survival. Analysis is limited to total gdT cells. Available database as Cancer Genome Atlas, and web server as Gene Expression Profiling Interactive Analysis 2 (GEPIA 2) could be used to assess if the Trm signature identified is associated with better or worse overall survival.
- Figure 5. Vg9Vd2 T-cells are selectively depleted, but can acquire tissue-residence, within HCC TILs. Distribution of Vd1 and Vd2 should be calculated among CD3+ T cells or total gd T cells to exclude bias due to the distribution of other lymphocytes populations. Clinical information of CRCLM patients are missing.
- Ref 40 (in line 84): the findings in that ref are not restricted to cirrhotic liver

Reviewer #1 (Remarks to the Author):

The Manuscript by Zakery N investigate the role of tissue resident gamma delta T cells in hepatocellular carcinoma, highlighting their possible use in cancer immunotherapy. The paper is quite nice, well written and correctly referenced. It reports experiments that are well designed, data which are convincing, and well interpreted. Some very minor revisions are nevertheless required about text and typo errors, which nevertheless do not affect the important significance of this study.

We thank the reviewer for their positive comments.

Reviewer #2 (Remarks to the Author):

1. In IHL, CD69+CD49a+ T cells included central memory (CD27+CD45RA-) and effector memory (CD27-CD45RA-) T cells (Supplementary Fig.1d). Did the author check the HCC TIL? For example, more effector memory or terminally differentiated $\gamma\delta$ T cells were detected?

We thank the reviewer for this question. We have now included additional data examining the memory phenotype (CD27 and CD45RA expression) on V δ 1 and V γ 9V δ 2 T-cells within HCC TILs (**new Fig.5b, new Supp.Fig5b**). In particular, we demonstrate that the V δ 1 T-cell subset within HCC TILs consists of a higher proportion of terminally differentiated effector memory (TEMRA, CD27-CD45RA+) cells in comparison to V γ 9V δ 2 T-cells (**new Fig.5b**),

suggesting that the V δ 1 T-cells may encounter greater antigen stimulation within HCC. A similar finding was observed for V δ 1 T-cells within paired background liver (**new Fig.5b**).

2. In Fig. 4, $\gamma\delta$ T-cell counts in the tumor was correlated with tumor size and patient prognosis. Which subset V δ 1 or V δ 2 T cells contribute to immunosurveillance dominantly? Though V γ 9V δ 2 T cells were enriched for expression of resident markers within HCC TILs, the frequency of V γ 9V δ 2 T cells was lower than that of V δ 1 T cells in the liver and tumor.

We thank the reviewer for this useful comment, and agree that a limitation of our immunohistochemistry (IHC) data was that we did not determine whether V δ 1 or V γ 9V δ 2 T-cells or both play dominant roles in natural immunosurveillance against HCC, although the reviewer correctly suggests that the correlation seen with pan- $\gamma\delta$ T cells is more likely to be attributable to the more numerous V δ 1 subset than the smaller V γ 9V δ 2 T-cell subset.

We have since attempted to perform further IHC staining to differentiate the contribution of V δ 1 and V δ 2 T-cells to patient survival within our cohort. However, while the pan- $\gamma\delta$ T cell antibody has been well-validated for IHC, there is a lack of V δ 1 and V δ 2 T-cell antibodies currently validated for this purpose. Despite our attempts at optimising protocols for V δ 1 and V δ 2 antibody immunostaining, we encountered a high degree of non-specific antibody binding, rendering results uninterpretable. In our second cohort of patients with HCC in whom we performed flow cytometry staining to distinguish V δ 1 and V δ 2 T-cell subsets within freshly isolated TILs, there were too few deaths by the time of study write-up to perform survival associations.

Therefore, as an alternative we have analysed a larger dataset from the Cancer Genome Atlas using the GEPIA2 web server (n=364 HCC). We find that higher expression of a $\gamma\delta$ -TCR gene expression signature (TRDC, TRGC1, TRGC2) in HCC is significantly associated with longer overall patient survival, supporting our immunostaining results. Comparing the gene signatures for V γ 9V δ 2 TCR (TRDC, TRGC1) and non-V γ 9V δ 2 $\gamma\delta$ TCR (TRDC, TRGC2), we see that both show significant associations with overall patient survival, with similar hazard ratios, suggesting they may both play a role in immunosurveillance within HCC (**new Supp.Fig.4f-g, new results lines 251-255**).

To further delineate the contribution of V δ 1 and V γ 9V δ 2 T-cells within HCC, we have now included additional data examining activation marker expression on V δ 1 and V γ 9V δ 2 T cells

directly isolated from HCC and paired background liver. We find that V δ 1 T-cells appear more activated within HCC, with higher *ex vivo* expression of the T cell activation markers HLA-DR, CD38 and CD25 within HCC TILs and liver, compared to V γ 9V δ 2 T-cells (**new Fig.5c, new Supp.Fig.5c, page 11**). The V δ 1 T cell population in HCC TILs also consisted of a higher proportion of terminally differentiated effector memory (CD27-CD45RA+) cells, suggesting greater antigen stimulation (**new Fig.5b**). We show V δ 1 T-cells have higher cytotoxic potential with higher Granzyme B expression in HCC TILs, although both V δ 1 and V γ 9V δ 2 T-cells isolated from HCC TILs maintain equivalent capabilities for IFN- γ production upon stimulation (**Fig.5d,e**). Overall, V δ 1 T-cells appear to encounter greater activation with higher cytotoxic potential within HCC TILs, which coupled with their higher frequency, may suggest a higher potential endogenous contribution to immunosurveillance within HCC compared to V γ 9V δ 2 T-cells.

We have accordingly revised the Results (**lines 290-299**) and Discussion (**lines 409-412**) to highlight this and to stress that regardless of whether they play a dominant physiological immunosurveillance role, we have concentrated on exploiting the presence of V γ 9V δ 2 T-cells for therapeutic targeting in view of their known ligands, capacity to acquire a T_{RM} phenotype within the liver and HCC, and ease of expansion.

3. For the strategy of intra-tumoral delivery of zoledronate combined with expanded V γ 9V δ 2 T cell therapy, systemic injection (Kobayashi et al, Cancer Immunol Immunother, 2011) or local (Intraperitoneal) injection (Wada et al, Cancer Med, 2014; ref. 52 in this manuscript) of zoledronate before V γ 9V δ 2 T cell infusion had been tried previously. But their efficacies seem to be limited. What do the authors think these results? I recommend the authors discuss this.

We thank the reviewer for raising this important point. As requested, we have now accordingly revised the Discussion (**lines 434-439, page 16**) to reference other studies, including the two studies mentioned above that have trialled systemic Zoledronic acid (ZOL) therapy or local ZOL injection in combination with V γ 9V δ 2 T-cells for cancer immunotherapy. We agree that the use of systemic ZOL therapy has previously demonstrated limited anti-tumour efficacy. We discuss that the high affinity of systemic ZOL for bone mineral and its rapid renal clearance may limit the systemic availability of ZOL for distant tumour cell uptake, likely accounting for the insufficient anti-tumour responses observed.

In contrast, local administration of ZOL has demonstrated greater promise with early signs of enhanced anti-tumour efficacy. In the small study of patients with malignant ascites by Wada *et al.* (**page 16**), intraperitoneal injections of ZOL combined with *ex-vivo* expanded V γ 9V δ 2 T-cells did reduce the tumour cell number in the ascites, although only 7 patients were included in the study all of whom had advanced malignancy. We also reference a study by Jarry *et al.* where intratumoural delivery of V γ 9V δ 2 T-cells, aminobisphosphonate and IL-2 in a murine xenotransplant model of glioblastoma led to significant tumour reduction; as well as *in vitro* studies demonstrating that ZOL treatment of cancer cell lines enhanced V γ 9V δ 2 T-cell mediated tumour cell lysis (**page 16, lines 437-439**). Our data adds further support to the proposal that intratumoural delivery of ZOL enhances IPP expression, increasing the local activation and anti-tumour function of V γ 9V δ 2 T-cells *in situ*.

4. In Fig. 3e, it was shown that IHL CD69+CD49a+ $\gamma\delta$ T cells expressed high PD-1. Do the authors think that the possibility of further combination of immune check point inhibitors with ZOL-based $\gamma\delta$ T cell therapy?

We thank the reviewer for this interesting question. Our data suggest that higher PD-1 expression on intrahepatic $\gamma\delta$ T-cells correlates with increased cell activation and cytokine production, rather than representing a functionally exhausted phenotype (**Fig.3f**). We have now also included data demonstrating that combining anti-PD-L1 blockade with $\gamma\delta$ -TCR stimulation of intrahepatic lymphocytes did not result in a significant increase in intrahepatic V γ 9V δ 2 IFN- γ or Granzyme B expression (**new Supp.Fig.3f; lines 226-228**).

In addition, we have combined anti-PD-L1 blockade within the co-culture of ZOL-treated HepG2 cells and ZOL-expanded V γ 9V δ 2 T-cells, and found no significant increase in the expanded V γ 9V δ 2 T-cells effector function (IFN- γ , TNF- α expression) (**new Fig.7h; lines 372-374**). Therefore our data suggest that anti-PD-1/PD-L1 receptor blockade may not provide additional benefit to ZOL- $\gamma\delta$ based therapy and we have accordingly revised the Results to mention this (**lines 226-228 page 10, 372-374 page 14**).

Reviewer #3 (Remarks to the Author):

Overall, I am not convinced that this manuscript combines its several findings to one

coherent story.

Major issues:

It would be helpful to focus on the Trm gd T cells in the liver and e.g. extend the Trm signature beyond and or confirm their signature by transcriptional data.

We thank the reviewer for this useful suggestion. We have now extended our description of tissue-resident gamma delta T cells in the liver, demonstrating that in addition to increased expression of the liver homing chemokine receptors CXCR6 and CXCR3, intrahepatic V δ 1 and V γ 9V δ 2 T_{RM} cells also demonstrate other features of classical tissue-residency, including expressing lower levels of the migratory chemokine receptor CX3CR1, and lack of CD62L required for secondary lymph node homing (**new Supp.Fig.1g-h**). We have also characterised the transcription factor profile of intrahepatic $\gamma\delta$ T_{RM} cells by intranuclear flow cytometry staining (**new Supp.Fig.1i**). We show that CD69+CD49a+ $\gamma\delta$ T_{RM} cells express significantly higher levels of the transcription factor Blimp-1 and lower levels of Eomes in comparison to CD69-CD49a- $\gamma\delta$ T cells within liver (**new Supp.Fig.1i**), consistent with the transcriptional profile associated with $\alpha\beta$ T_{RM} cells. Tcf-1 also demonstrates a trend for higher expression on CD69+CD49a+ $\gamma\delta$ T_{RM} (**new Supp.Fig.1i**), in line with their capacity for longevity or self-renewal we demonstrate in the allograft setting (lines 186-188).

If the authors want to make the point that activation of V γ 9V δ 2 cells leads to liver residency, this should be shown, at least by analysis of their liver homing capacity

We thank the reviewer for this useful comment. We have now included additional data showing that 7-day TCR stimulation of V γ 9V δ 2 T-cells within PBMCs (using a plate-bound anti- $\gamma\delta$ TCR antibody stimulus), leads to increased activation of V γ 9V δ 2 T-cells demonstrated by increased CD38 expression (**new.Supp.Fig.7d**), accompanied by a time-dependent induction of *de novo* CD69⁺CD49a⁺ or CD69⁺CD103⁺ tissue-residency marker expression (**new.Supp.Fig7d-e**), and a significant increase in expression of the liver homing chemokine receptors CXCR6 and CXCR3 (**new Supp.Fig.7f**). We show that CD103 and CD49a expression is significantly higher on the activated CD38^{high} expressing V γ 9V δ 2 T-cells compared to CD38^{low} V γ 9V δ 2 T-cells following 7-day TCR stimulation (**new Supp.Fig.7d**). Furthermore, we show that repeated V γ 9V δ 2 T-cell stimulation is required (7-days), rather than short-term (4hour- or 24-hour) stimulation, for induction of a *de novo* tissue-residency phenotype (**new Supp.Fig7e**). Exposure to IL-2 over 6-days induced a lower level of CD49a

expression, whereas CD103 remained undetectable on V γ 9V δ 2 T-cells without additional TCR stimulation (**new Supp.Fig7d,e**). We have accordingly revised the Results to include this additional data (**lines 352-360**).

Minor issues:

- Figure 1. Compartmentalization of V δ 1 and V γ 9V δ 2 T-cells with a tissue-resident phenotype in human liver. Are the different levels of expression of CD49a and CD103 associated with age / sex or any other clinical parameter of patients?

Thank you for this question. The level of CD69⁺CD103⁺ expression on intrahepatic V δ 1 and V γ 9V δ 2 T-cells demonstrated a weak inverse correlation with patient age (**new.Supp.Fig.1f**). No significant correlation was observed between the level of CD69⁺CD49a⁺ expression on V δ 1 or V γ 9V δ 2 T-cells with patient age (**new.Supp.Fig.1f**). No sex related differences in CD69⁺CD103⁺ or CD69⁺CD49a⁺ expression on V δ 1 or V γ 9V δ 2 T_{RM} were observed (**new.Supp.Fig.1f**). We have now made reference to this within the Results (**lines 134-135**).

- Fig 1b: it seems like V δ 1⁺ and V δ 1⁻ are not separated

We have amended the flow plot to demonstrate more clearly that V δ 1 and V δ 2 T-cells were separated for the analysis in **new.Fig.1c** (previously Fig1b).

- The pan gd TCR staining in Fig 2c,d is not convincing, please show full gating strategies or at least all T cells or all lymphocytes in a gdTCR vs. HLA-A2 plot.

As requested, we have now included full gating strategies demonstrating the pan- $\gamma\delta$ T-cell gating (**new Fig.2c, Supp.Fig.2a**), as well as all T cells in a $\gamma\delta$ TCR vs. HLA-A2 plot (**new Supp.Fig.2a**).

- Figure 2. Long-lived hepatic retention and replenishment of V δ 1 and V γ 9V δ 2-TRM: fig 2a analysis limited to 3 patients, hard to draw conclusion considering variability among patients.

We have added an additional set of peripheral, hepatic and portal venous blood samples from a patient undergoing a TIPS procedure to **Fig.2b**. These samples are challenging to acquire due to the uncommon and specialist nature of the TIPS procedure and Covid restrictions in place. Nevertheless, all samples from the four patients are highly consistent in demonstrating a complete absence of CD69+CD49a+ and CD69+CD103+ $\gamma\delta T_{RM}$ cells in peripheral, hepatic and portal venous blood (**new.Fig.2b**). We have also added a sentence acknowledging the limitation of small patient numbers for TIPS and HLA-mismatched allografts due to scarcity of these valuable samples (**line 205**).

Fig 2d should show data on gated vd1 and Vd2 rather than pan-gd. Since it's only two patients the data could be summarized with dimensional reduction analysis as U-MAP or t-SNE.

We thank the reviewer for this suggestion. We have now shown T_{RM} marker expression gating on V δ 1 and V δ 2 T-cell subsets for the two patients (**new Supp.Fig2c-d**). Variation in the panels used for the redo liver transplant patients means that we cannot unfortunately summarise the data with a t-SNE or U-MAP, however we have clarified the gating strategy in full now, and expanded on the T_{RM} phenotyping as described above. In addition, we have applied the excellent suggestion of t-SNE analysis to our initial characterisation of hepatic $\gamma\delta$ - T_{RM} , where we had been able to use consistent antibody panels, now included as **new Fig.1b**.

- Figure 3. Distinct functional profile of hepatic gdTRM. Fig. 3e plot showing PD1 expression among CD69+CD49a+ Vd2 cell should be representative of statistical mean.

Thank you for pointing out this error. We have now amended the PD-1 expression plot to show an example more representative of the statistical mean (**new.Fig.3e**).

- Figure 4. gd T-cell counts in HCC are associated with tumour size and patient survival. Analysis is limited to total gdT cells. Available database as Cancer Genome Atlas, and web server as Gene Expression Profiling Interactive Analysis 2 (GEPIA 2) could be used to assess if the Trm signature identified is associated with better or worse overall survival.

We thank the reviewer for this very useful suggestion. Interrogating the GEPIA2 web server (incorporating 364 HCC tumours from the Cancer Genome Atlas) for $\gamma\delta$ -TCR gene expression signature (TRDC, TRGC1, TRGC2), we find a similar significant association of total $\gamma\delta$ T cell counts with overall patient survival (**new Supp.Fig.4f**) to the one we had found in our dataset (**Fig.4d**). We have also included data comparing the gene signatures for the V γ 9V δ 2 (TRDC, TRGC1) and non-V γ 9V δ 2 $\gamma\delta$ TCR (TRDC, TRGC2), both showing significant associations with similar hazard ratios, suggesting they may both play a role in immunosurveillance (**new Supp.Fig.4g**). Furthermore, a combined CD69⁺CD103⁺ and $\gamma\delta$ -TCR gene signature showed a favourable prognostic association with overall survival in HCC, not detected with an equivalent CD69⁺CD103⁺ $\alpha\beta$ CD8⁺TCR gene signature (**new.Supp.Fig.5e**). A combined CD69⁺CD49a⁺ and $\gamma\delta$ TCR gene signature did not show any significant difference with overall survival (**new Supp.Fig.5e**). This suggests a possible prognostic association of CD69⁺CD103⁺ $\gamma\delta$ T-cells, however we are aware of the limitations since the T_{RM} gene signature may be contributed to by other cells (not only $\gamma\delta$ T-cells), therefore we have now alluded to this within the manuscript whilst cautioning that we cannot make definitive conclusions (**lines 283-288**).

- Figure 5. V γ 9V δ 2 T-cells are selectively depleted, but can acquire tissue-residence, within HCC TILs. Distribution of V δ 1 and V δ 2 should be calculated among CD3⁺ T cells or total $\gamma\delta$ T cells to exclude bias due to the distribution of other lymphocytes populations. Clinical information of CRCLM patients are missing.

Thank you for this suggestion. We have now calculated the frequency of V δ 1 and V γ 9V δ 2 T-cells as a percentage of total CD3⁺ T cells (rather than CD45⁺ lymphocytes) (**new Fig.5a**). Importantly, we see similar findings with a reduced frequency of V γ 9V δ 2 T-cells in the blood, liver and tumours of patients with HCC in comparison to CRCLM (**new Fig.5a, lines 263-264**). In addition, we have provided clinical information of the patients with CRCLM in our cohort (**new Supp.Table4**).

- Ref 40 (in line 84): the findings in that ref are not restricted to cirrhotic liver

Thank you for pointing out this error, we have now corrected this sentence in the manuscript (**line 84**).

We hope that with the changes detailed above, the manuscript will now be suitable for publication in Nature Communications. All authors concur with these additions, which are highlighted throughout the manuscript.

Thank you very much for your consideration.

Yours sincerely,

Mala Maini PhD FRCP FMedSci

Professor and Honorary Consultant, Viral Immunology
Wellcome Trust Senior Investigator

REVIEWERS' COMMENTS

Reviewer #1 (Remarks to the Author):

The authors have exhaustively improved the quality and the significance of their results, moreover they have added some Supplementary Figures that clarify the analysis. In the new form, the manuscript is acceptable for the publication considering their potential implications in therapy.

Reviewer #2 (Remarks to the Author):

I thank the authors for their great effort in addressing my comments. I am satisfied by their responses.

Reviewer #3 (Remarks to the Author):

The authors made a serious effort to address all the issues raised by the two critical reviewers. The ms should now be suitable for publication in ncomms. However, I am not sure about the policy of ncomms and the funding Wellcome Trust regarding the data availability statement line 632.

Reviewer #1 (Remarks to the Author):

The authors have exhaustively improved the quality and the significance of their results, moreover they have added some Supplementary Figures that clarify the analysis. In the new form, the manuscript is acceptable for the publication considering their potential implications in therapy.

We thank the reviewer for their positive comments and for recognising the impact of the study.

Reviewer #2 (Remarks to the Author):

I thank the authors for their great effort in addressing my comments. I am satisfied by their responses.

We thank the reviewer for their favourable comments.

Reviewer #3 (Remarks to the Author):

The authors made a serious effort to address all the issues raised by the two critical reviewers. The ms should now be suitable for publication in ncomms.

However, I am not sure about the policy of ncomms and the funding Wellcome Trust regarding the data availability statement line 632.

We thank their reviewer for their positive comments. We have now provided all raw data generated in the study within the Source Data File and have amended the data availability statement accordingly.

We hope that with the changes detailed above, the manuscript will now be suitable for publication in Nature Communications. All authors concur with these additions, which are highlighted throughout the manuscript.

Thank you very much for your consideration.

Yours sincerely,

Mala Maini PhD FRCP FMedSci
Professor and Honorary Consultant, Viral Immunology
Wellcome Trust Senior Investigator